# Identifying the Multi-Scale Influences of Climate Factors on Runoff Changes in a Typical Karst Watershed Using Wavelet Analysis

**Luhua Wu** [1,2,3], **Shijie Wang** [2,3], **Xiaoyong Bai** [2,3,*], **Fei Chen** [2,3], **Chaojun Li** [2,3], **Chen Ran** [2,3] and **Sirui Zhang** [2,3]

1   School of Economics and Management, Tongren University, Tongren 554300, China
2   State Key Laboratory of Environmental Geochemistry of Geochemistry, Institute of Geochemistry, Chinese Academy of Sciences, Guiyang 550081, China
3   Puding Karst Ecosystem Observation and Research Station, Chinese Academy of Sciences, Puding 562100, China
*   Correspondence: baixiaoyong@vip.skleg.cn

**Abstract:** Identifying the impacts of climatic factors on runoff change has become a central topic in climate and hydrology research. This issue, however, has received minimal attention in karst watersheds worldwide. Multi-resolution analysis (MRA), continuous wavelet transform (CWT), cross wavelet transform (XWT) and wavelet transform coherence (WTC) are used to study the teleconnection in time and frequency between climate change and hydrological processes in a typical karst watershed at different time scales. The main results are: (1) All climatic factors exhibit a main cycle at 12-month time scales with runoff changes, but the main periodic bandwidth of rainfall on runoff changes is much wider than that of temperature and evaporation, indicating that rainfall is the main factor affecting runoff changes. (2) In other cycles, the impact of rainfall on runoff changes is the interlacing phenomena with positive and negative, but the impact of temperature and evaporation on runoff change is mainly negative. (3) The response of runoff to rainfall is in time in the high-energy region and the low-energy significant-correlation region and has shown a positive correlation with a smaller phase angle, but it is slightly lagged at 16-month time scales. Moreover, the runoff change lags behind temperature and evaporation for 1–2 months in those regions. (4) It has been found that there is a strong effect of rainfall over runoff, but a lesser effect of temperature and evaporation over runoff. The study sheds light on the main teleconnections between rainfall, evapotranspiration and surface runoff, which in turn might help to attain the better management of water resources in typical karst watersheds.

**Keywords:** karst; watershed; runoff change; climate factors; wavelet analysis

## 1. Introduction

The changes in hydrology and water resources caused by climate change have stimulated hydrologists to pay attention to and study the impacts in this field, which have become one of the popular issues at home and abroad [1–3]. Runoff change is mainly affected by climate change and underlying surface conditions [4–6]. The most important manifestations of climate change on runoff are the changes in rainfall amount and temporal and spatial distributions [7]. Climatic and hydrological processes have highly non-linear and unstable characteristics due to the complex exchange process of the Earth's atmosphere system [8–10] and the difference in watersheds' geographical and human environment characteristics; these characteristics cause great difficulties in the simulation and prediction of hydrological changes. Therefore, studying the multi-scale evolution law in the interaction process between meteorology and hydrology has scientific significance for the management and optimal regulation of water resources in river watersheds.

Most of the previous studies have focused on the correlation between meteorological factors and runoff, but limited analysis has been conducted on the multi-temporal characteristics of runoff and major meteorological elements [6,11–17]. In addition, the traditional hydrological research methods, such as linear correlation, the Pearson correlation coefficient, the linear trend method and the multiple regression method, can only reveal the variation characteristics on a single time scale. The evolution relationship and interaction characteristics of the two hydrometeorological variables on multi-time-scales are impossible to demonstrate. Spectral analysis and spatial coherency can reveal the scale-dependent relationships between the variables but are only applicable to stationary systems. Hydrometeorological series behavior belongs to non-stationary systems [18]. Wavelet analysis can deal with non-stationary data series and thereby provide an opportunity to analyze the temporal patterns of hydrometeorological series at multiple scales.

The climatic factor is the driving factor of runoff change; therefore, different time scale correlations must exist between runoff and the climatic factor in its oscillation frequency. Wavelet analysis, especially cross wavelet analysis, had been gradually applied to the analysis and study of runoff changes and meteorological factors in river watersheds on multi-time-scales in recent years [19–21] and had also been used to determine the overall and scale-dependent similarities of the temporal patterns of soil moisture [22]. Multiresolution analysis (MRA) can study signals represented at different resolutions [23]. This method can be used to decompose a signal into a progression of successive approximations and details in increasing order of resolution [24]. Continuous wavelet transform (CWT) is a common tool for analyzing localized intermittent oscillations in time series, and it is often desirable to examine two time series together that may be expected to be linked in some way. We can show the strength of sequence signals at different time scales by analyzing the wavelet power spectrum. cross wavelet transform (XWT) will expose their common power and relative phase in time–frequency space and can reflect that two sequences have the same energy spectrum region after wavelet transform, thus revealing the significance of the interaction between the two sequences in different time–frequency domains. Wavelet transform coherence (WTC) can find significant coherence although the common power is low [19]. In this field, CWT has been recently used to determine the effect of climatic phenomena on stream flow regimes [25–28], runoff processes [29–31], surface–groundwater interactions and the hydrogeological behavior of karst systems [32,33]. Furthermore, XWT, which has strong signal coupling and resolution ability, can show the common high-energy region and phase correlation of two time-series data. However, XWT has a great unsolved shortcoming that cannot find significant coherence when analyzing the low-energy regions of two time series' data in the time–frequency domain, and its functional defects in the low-energy areas must be compensated for by WTC [34]. In view of this, the coupling of MRA, CWT, XWT and WTC will be generally applied in the field of hydrometeorology.

At present, the implications of climate and anthropic pressures on the short- to long-term changes in the water resources of a Mediterranean karst were assessed by using wavelet analysis [23], and the non-stationary relationships of ocean and atmosphere mean conditions and freshwater discharge, which were integrated at the continental scale, were studied by using XWT [27]. In addition, the impacts of rainfall, air temperature and evapotranspiration on the annual runoff in the source region of the Yangtze River were investigated in the time domain by using wavelet analysis and multiple regression [17]. WTC was used to determine the overall and scale-dependent similarities of the temporal patterns of soil moisture in the karst catchments of Southwestern China [32]. CWT analysis was used to detect the trends and periodicity in sediment discharge, whilst WTC was used to detect the temporal covariance between sediment discharge and water discharge, rainfall, potential evapotranspiration and vegetation index in two typical karst watersheds in southwest China [33] and to assess the relative importance of catchment properties in modulating streamflow and modes of variability in West Africa and Central Africa [35].

Karst landforms are developed in highly heterogeneous carbonate rocks that are easily eroded by flowing water, widely distributed in Southwest China and generally have differ-

ent hydrogeological characteristics from non-karst areas [36]. Thus, the karst watersheds are characterized by broken surfaces, low runoff coefficients, serious underground leakage, thin surface soil and poor regulation and water conservation capacities [37–40]. The unique two-dimensional/three-dimensional hydrogeological structure can accelerate the hydrological process [41,42]. In particular, rainfall drains rapidly to underground systems through numerous cracks and fissures [40,43–46]. The soil–epikarst system plays important roles in runoff generation due to the large storage capacity and high infiltration rate of karst carbonate fissures and fractures [2]; consequently, runoff changes in karst areas are sensitive to climatic factors, and small climatic fluctuations will cause large fluctuations in runoff. The appearance, storage and circulation of water in karst aquifers are apparently different from those of water in non-karst areas. The special hydrological process in karst areas will lead to the influence of climate change on runoff with time lag or advance at different time scales. In addition, the hydrometeorological evolution in karst areas has obvious seasonal and multi-scale characteristics. Significant differences exist in the evolution and influence relationships at different scales, especially the intrinsic relationship between monthly and seasonal rainfall, evaporation, temperature and runoff; their vibration energy distribution characteristics and correlations in time and frequency domains are extremely complex. Numerous studies have focused solely on non-karst watersheds. On the contrary, the impacts of climatic factors on runoff (surface runoff) changes have rarely been identified for karst watersheds. Specifically, the research on the time-varying characteristics of climate and runoff and their coupling relationship in karst trough valley watersheds is scarce. Therefore, the objectives of this study are to (1) analyze the multiscale temporal variability effects of runoff with climatic factors, (2) characterize the coupling relationship between runoff and climatic factors in common high- and low-energy regions and high-correlation regions at different time scales and (3) provide a theoretical basis and technological support for water resource safety management in karst watersheds.

## 2. Study Site

The Yinjiang River watershed (108°21′21″–108°47′27″ E, 27°53′17″–28°13′57″ N), which is located in northeast of Guizhou Province (Figure 1a), is a typical karst watershed of a trough valley, SW China. It covers an area of 691.56 km$^2$, with the karst area of 376.77 km$^2$ and non-carbonate rocks area of 314.79 km$^2$, accounting for 54.68% and 45.32% of the total watershed area, respectively. Elevation in the study area decreases from southeast to northwest, ranging in a large scope with an elevation range of 439–2466 m above sea level and a mean elevation of 1033 m above sea level (Figure 1b).

The southeast part of the watershed is dominated by non-karst areas, and the karst is widely distributed in the middle and northwest parts of the watershed. A small number of banded non-karst regions are concentrated in the western, central and northern parts of the watershed. Six types of lithology are present, namely, homogenous limestone, interbedded limestone and clastic rock, clastic rock of limestone interlayer, non-carbonatite, homogenous dolomite, mixture of homogeneous limestone and dolomite (Figure 1c). A karst valley with a geographical background of a syncline structure in the center of the valley with steep bedding slopes exists on both sides. The land surface is steep and broken with numerous underground cracks, causing a severe underground loss of rainfall and runoff. The middle part of the watershed is a typical deep-cut karst trough valley, and the middle part of the trough is a karst valley with a synclinal structure as its geological background. Both sides of the trough are steep beddings or inversion slopes, and the top of the trough is over 1000 m above sea level; thus, it has a good ecological three-dimensional climate characteristic.

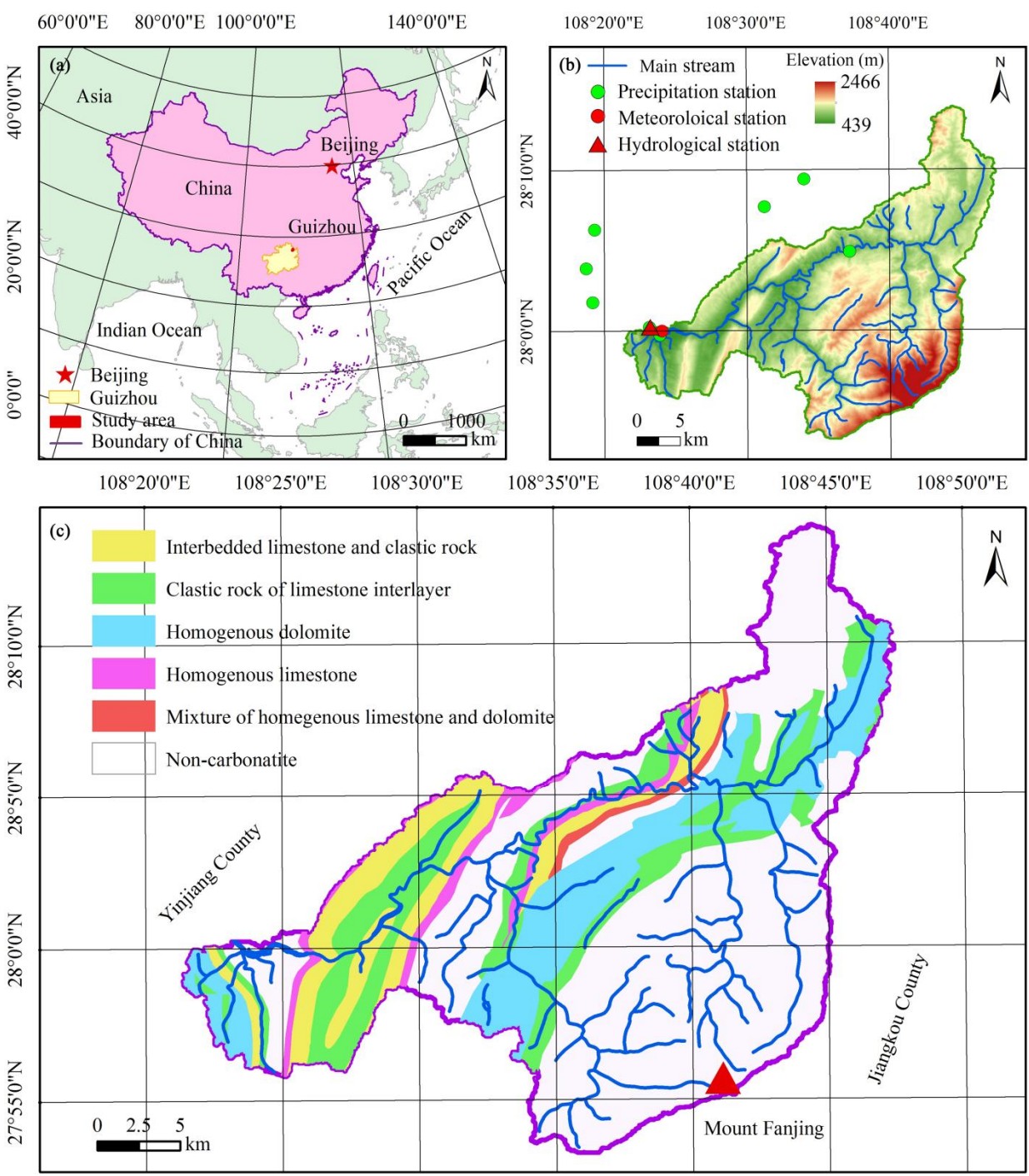

**Figure 1.** Location and overview of the study area. Study area location in China (**a**), topography (**b**) and lithology (**c**) in Yinjiang River watershed.

## 3. Materials

The monthly rainfall (P) data of 8 rainfall observation stations and the monthly runoff (Q) data (long data from January 1984 to December 2015) at the hydrological station were collected from the Guizhou Provincial Hydrology and Water Resources Bureau (http://www.gzswj.gov.cn/hydrology_gz_new/index.phtml) (accessed on 15 September 2016). The monthly evaporation (E) data (January 1984 to December 2015), which were also derived from the Guizhou Provincial Hydrology and Water Resources Bureau, were measured by the evaporation dish of the hydrological observation station and represented the water evaporation of the water surface or soil. The monthly temperature (T) data

of the weather station corresponding to the runoff time series were obtained from the China Meteorological Data Sharing Service System (http://cdc.cma.gov.cn/) (accessed on 5 August 2021). The average annual rainfall data for the watershed were interpolated by the Kriging method with the annual rainfall data of all 8 rainfall stations (both inside and outside the watershed). The DEM data with a spatial resolution of 30 m were obtained from the International Scientific and Technical Data Mirror Site, Computer Network Information Center, Chinese Academy of Sciences, which could be downloaded from the Geospatial Data Cloud (http://www.gscloud.cn) (accessed on 12 September 2021). The lithology data were derived from the Karst Scientific Data Center (http://www.karstdata.cn/) (accessed on 12 September 2021), Institute of Geochemistry, Chinese Academy of Sciences.

## 4. Methodology

A series of wavelet analysis methods were used to identify the multi-scale impact of climate factors on runoff change in the Yinjiang River watershed. These wavelet analysis methods included MRA [47], CWT [19–21,48,49], XWT [19–21], cross wavelet phase angle (CWPA) [50] and WTC [19]. MRA was carried out using a free MATLAB software package provided by the WaveLab Development Team and available at http://statweb.stanford.edu/~wavelab/, (accessed on 12 September 2021). Other methods, including CWT, XWT, WTC, and CWPA, were carried out using a free MATLAB software package (Mathworks, Natick, MA, USA) kindly provided by Grinsted et al. [19] at http://noc.ac.uk/using-science/crosswavelet-wavelet-coherence, (accessed on 12 September 2021). The package includes code originally written by Torrence and Compo [20] of the University of Alaska, available at http://paos.colorado.edu/research/wavelets/, (accessed on 12 September 2021). The flowchart for identifying the multi-scale influences of climate factors on runoff changes is shown in Figure 2.

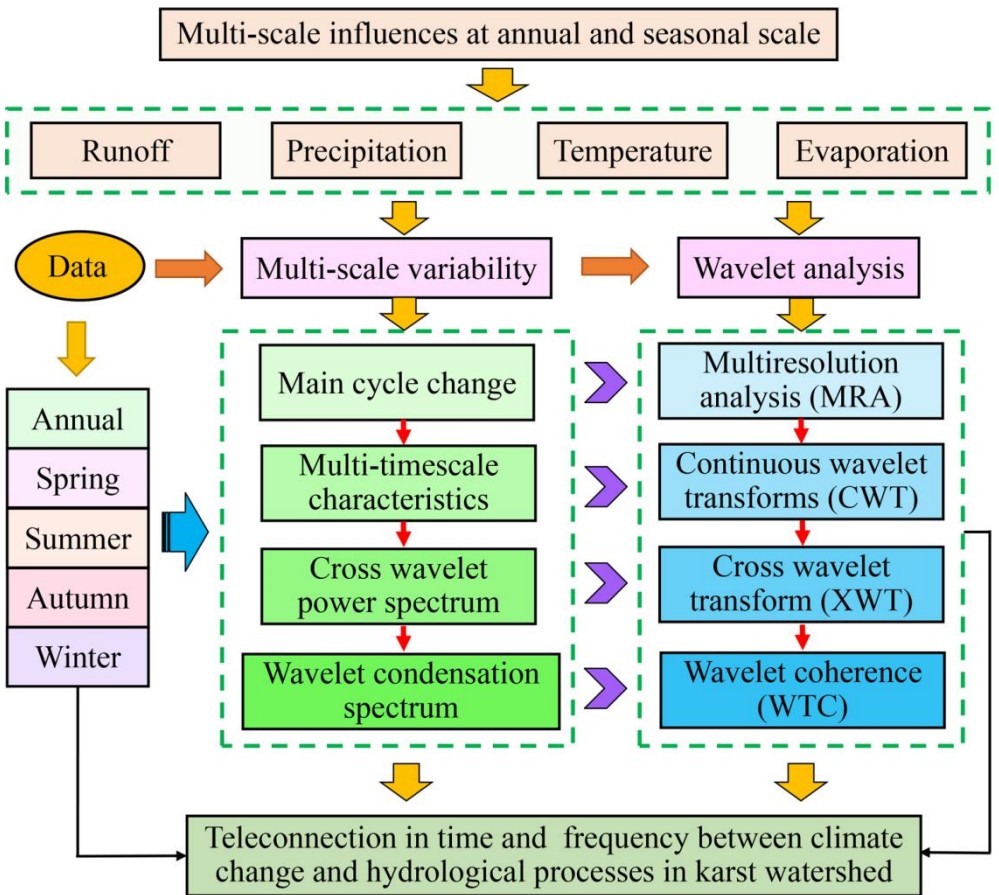

**Figure 2.** Flowchart for identifying the multi-scale influences of climate factors on runoff changes.

### 4.1. MRA

Choosing the particular values of $a_0 = 2$ and $s_0 = 1$ corresponds to the dyadic case used in MRA. The aim is to reduce/increase the resolution by a factor of 2 between two scales. Therefore, the approximation of a signal $x(t)$ at a resolution $j$, denoted by $A_x^j$, and the detail of the same function at a resolution $j$, denoted by $D_x^j$, are defined by:

$$A_x^j(t) = \sum_{k=-\infty}^{+\infty} C_{j,k} \psi_{j,k}(t) \tag{1}$$

$$D_x^j(t) = \sum_{k=-\infty}^{+\infty} D_{j,k} \Phi_{j,k}(t) \tag{2}$$

where $\Phi_{j,k}(t)$ is a scaled and translated basis function called the scaling function [47], which is determined with $\psi_{j,k}(t)$ when a wavelet is selected. $C_{j,k}$ is the scaling coefficient given the discrete sampled values of $x(t)$ at resolution $j$ and location $k$. It is calculated from $\Phi_{j,k}(t)$ in a similar way for the wavelet coefficient $D_{j,k}$ from $\psi_{j,k}(t)$ for detailed mathematical expressions [49].

The signal $x(t)$ can be reconstructed from the approximation and detail components as:

$$x(t) = A_x^j(t) + \sum_{j=1}^{J} D_x^j \tag{3}$$

where $J$ is the highest resolution level considered. Since the MRA ensures variance is well captured in a limited number of resolution levels, the analysis of energy distribution in the sampling time series across scales gives a good idea of the energy distribution across frequencies.

### 4.2. CWT

The wavelet transform can be seen as a bandpass filter of uniform shape and varying location and width [20]. The continuous wavelet transform (CWT) $W_x(a, \tau)$ of a time series $x(t)$ is given as follows:

$$W_x(a, \tau) = \int_{-\infty}^{+\infty} x(t) \psi^*(t; a, \tau) d_t \tag{4}$$

$$\psi(t; a, \tau) = \frac{1}{\sqrt{a}} \psi\left(\frac{t - \tau}{a}\right) \tag{5}$$

$$\psi(t) = \sqrt[4]{\frac{1}{\pi}} \cos(kt) e^{-\frac{t^2}{2}} \tag{6}$$

where $W_x(a, \tau)$ represents a group of wavelet functions, $W_{(a,b)}$, based on a mother wavelet $\psi$ which can be scaled and translated, modifying the scale parameter $a$ and the translation parameter $\tau$, respectively. $\psi^*(t; a, \tau)$ corresponds to the complex conjugate of $\psi(t; a, \tau)$. $\psi(t)$ is the Morlet wavelet function, $k$ is the non-dimensional frequency, here taken to be 6 to satisfy the admissibility condition, and $t$ is time.

The statistical significance of wavelet power can be assessed relative to the null hypothesis that the signal is generated by a stationary process with a given background power spectrum ($P_\eta$). Many geophysical time series have distinctive red noise characteristics that can be modeled very well by a first-order autoregressive (AR1) process. The Fourier power spectrum of an AR1 process with lag-1 autocorrelation $\alpha$ [48] is given by

$$P_\eta = \frac{1 - \alpha^2}{\left|1 - \alpha e^{-2i\pi\eta}\right|^2} \tag{7}$$

The probability that the wavelet power of a process with a given power spectrum $(P_\eta)$ [19] is greater than $p$ is

$$D\left(\frac{\left|W_n^X(s)\right|^2}{\sigma_X^2} < p\right) = \frac{1}{2}P_\eta \chi_v^2(p) \tag{8}$$

where $\eta$ is the Fourier frequency index. $v$ is equal to 1 for real and 2 for complex wavelets.

### 4.3. XWT

The XWT of two time series $X$ and $Y$ is defined as

$$W^{XY} = W^X W^{Y*} \tag{9}$$

where * denotes complex conjugation. We further define the cross wavelet power as $|W^{XY}|$. The complex argument $\arg(W^{XY})$ can be interpreted as the local relative phase between $X$ and $Y$ in time–frequency space. The theoretical distribution of the cross wavelet power of two time series with background power spectra $P_k^X$ and $P_k^Y$ is given as

$$D\left(\frac{W_n^X(s)W_n^Y(s)}{\sigma_X \sigma_Y} < p\right) = \frac{Z_v(p)}{v}\sqrt{P_k^X P_k^Y} \tag{10}$$

where $Z_v(p)$ is the confidence level associated with the probability $p$ defined by the square root of the product of two $X^2$ distributions [19].

We use the circular mean of the phase over regions with higher than 5% statistical significance that are outside the cone of influence (COI) to quantify the phase relationship. This is a useful and general method for calculating the mean phase. The circular mean of a set of angles $(a_i, i = 1 \ldots n)$ is defined as

$$a_m = \arg(X, Y) \; with \; X = \sum_{i=1}^n \cos(a_i) \; and \; Y = \sum_{i=1}^n \sin(a_i) \tag{11}$$

It is difficult to calculate the confidence interval of the mean angle reliably since the phase angles are not independent. The number of angles used in the calculation can be set arbitrarily high simply by increasing the scale resolution. However, it is interesting to know the scatter of angles around the mean. For this, we define the circular standard deviation as

$$S = \sqrt{-2\ln(R/n)} \tag{12}$$

where $R = \sqrt{X^2 + Y^2}$. The circular standard deviation is analogous to the linear standard deviation in that it varies from zero to infinity.

### 4.4. WTC

Cross wavelet power reveals areas with high common power. Another useful measure is how coherent the cross wavelet transform is in time–frequency space. We define the wavelet coherence of two time series as

$$R_n^2(s) = \frac{\left|S(s^{-1}W_n^{XY}(s))\right|^2}{S(s^{-1}|W_n^X(s)|^2) \cdot S(s^{-1}|W_n^Y(s)|^2)} \tag{13}$$

where $S$ is a smoothing operator. Notice that this definition closely resembles that of a traditional correlation coefficient, and it is useful to think of the wavelet coherence as a localized correlation coefficient in time–frequency space. We write the smoothing operator $S$ as

$$S(W) = S_{scale}(S_{time}(W_n(s))) \tag{14}$$

where *S* scale denotes smoothing along the wavelet scale axis and *S* time denotes smoothing in time. It is natural to design the smoothing operator so that it has a similar footprint as the wavelet used. For the Morlet wavelet, a suitable smoothing operator is given by Torrence and Compo [20]:

$$S_{time}(W)|_s = \left( W_n(s) \cdot c_1^{\frac{-t^2}{2s^2}} \right) \Big|_s \tag{15}$$

$$S_{time}(W)|_s = (W_n(s) \cdot c_2 \, \Pi(0.6s))|_n \tag{16}$$

where $c_1$ and $c_2$ are normalization constants and $\Pi$ is the rectangle function. The factor of 0.6 is the empirically determined scale decorrelation length for the Morlet wavelet [20]. In practice, both convolutions are conducted discretely and therefore the normalization coefficients are determined numerically.

In this study, the cross wavelet energy, wavelet correlation agglomeration and phase spectra were calculated for monthly temperature, evaporation and rainfall and monthly runoff series to analyze the multi-temporal correlation amongst temperature, evaporation, rainfall and runoff. The hydrometeorological variables were used as input and output signals to characterize the responses of runoff changes to climatic factors in the Yinjiang River watershed. Climatic factors (rainfall, temperature and evaporation) were taken as input signals, and runoff was taken as an output signal. The correlation between climatic factors and runoff signals in the frequency and time domains at different energies was analyzed by using XWT and WTC. We had focused our analysis on the P–Q relationship to assess the impact of rainfall on runoff changes. Besides that, the E–Q and T–Q relationship were used to assess the impacts of evaporation and temperature on runoff changes, respectively.

## 5. Result Analysis

### 5.1. Annual, Seasonal and Monthly Evolution of Runoff and Climatic Factors on Main Scales Analyzed by MRA

#### 5.1.1. Monthly Variation

The main aim of this section is to visualize the distribution of energy across scales (or resolution levels) of the hydrogeological time series. The MRA was performed on monthly data, and the results for the first 10 MRA levels are shown in Figure 3. Overall, the energy is distributed variably across levels in the hydrogeological time series and has significant periodic characteristics in different time scales, especially at a large scale. Runoff and rainfall show high energy oscillation at all levels, but they fluctuate with time, which indicates that the high-energy differences in monthly runoff and rainfall explain most of the differences. The energy vibrations at all time scales have a high consistency, and the vibration consistency is significant at a large scale, which demonstrates that runoff is significantly affected by rainfall. Evaporation shows high-energy vibrations at all time scales. Evaporation oscillation that gradually diminishes at 16-, 32- and 64-month time scales, however, is consistent with runoff and rainfall at 128- and 256-month time scales. Several obvious abrupt changes are detected from temperature in the vibration characteristics of 1 month to 4 months with the mutation years of 1990 (70-month) and 2000 (200-month) according to energy distribution. The oscillation characteristics of temperature are consistent with those of runoff and rainfall at time scales of 16, 64, 128 and 256 months. After 128-month time scales, the oscillation characteristics of runoff, rainfall, evaporation and temperature all appear to have the same vibration characteristics at 10–20-year time scales.

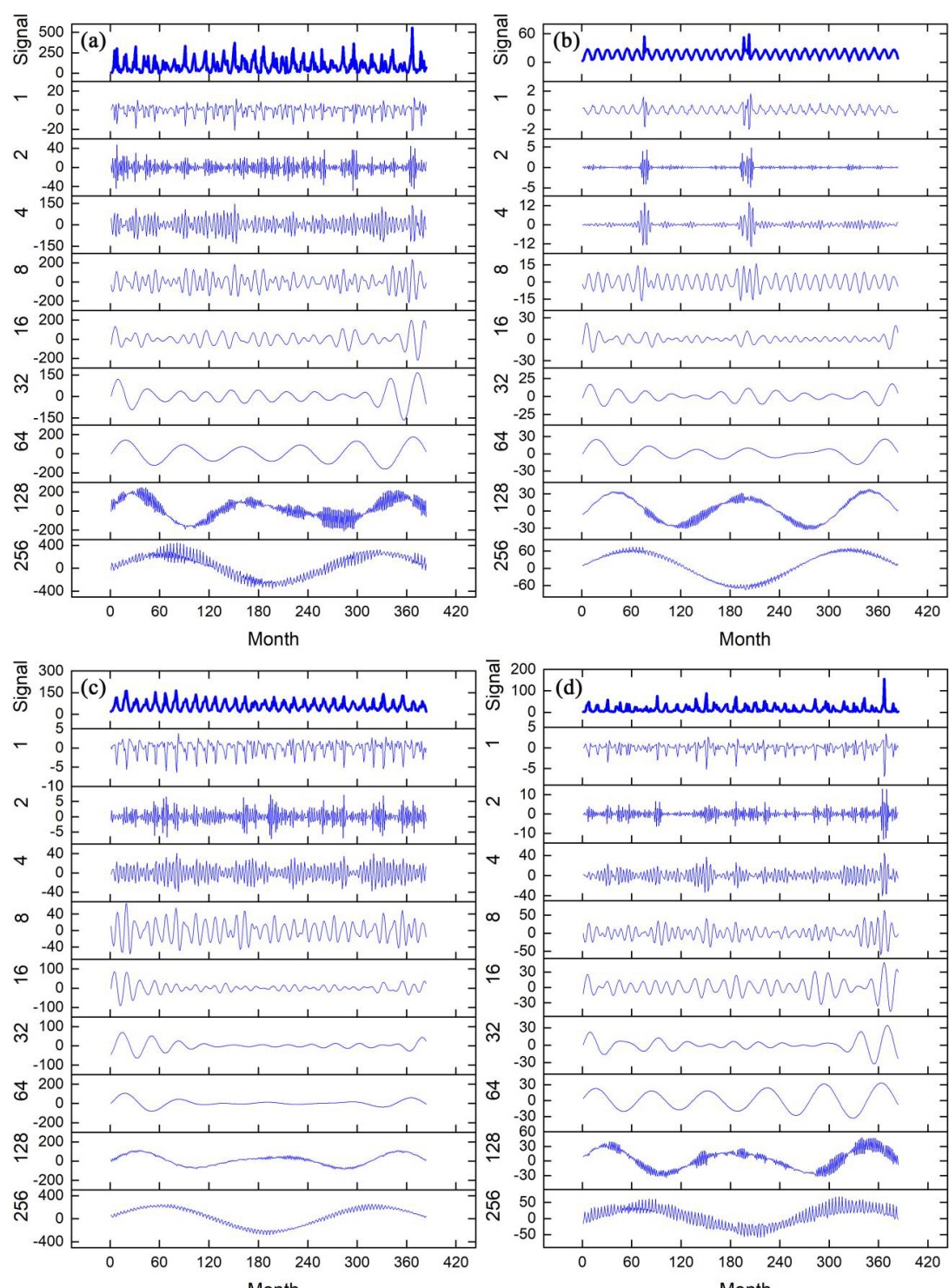

**Figure 3.** Multi-time-scale variations of monthly rainfall (**a**), temperature (**b**), evaporation (**c**) and runoff (**d**).

5.1.2. Seasonal Variation

As shown in Figures 4–7, there are some synchronization characteristics on different time scales for the evolution characteristics of surface runoff and climate factors on the annual scale and four seasonal scales after scale segmentation by MRA. Except for the small difference in runoff in autumn and winter, the evolution characteristics of runoff at other scales are basically synchronous, especially in summer. The amplitude of rainfall oscillation

at each time scale is larger than that of runoff, but a minimal difference exists between them in spring, autumn and winter. Evaporation is asynchronous with temperature for the variation characteristics at 1-year scale in spring and winter and at less than 4-year scale in summer but relatively synchronous with insignificantly different amplitudes at other scales. Over 16-year time scales, the time series of each factor has shown the consistent evolution characteristics in four seasons and the amplitude of each factor increases with the increase in time scale.

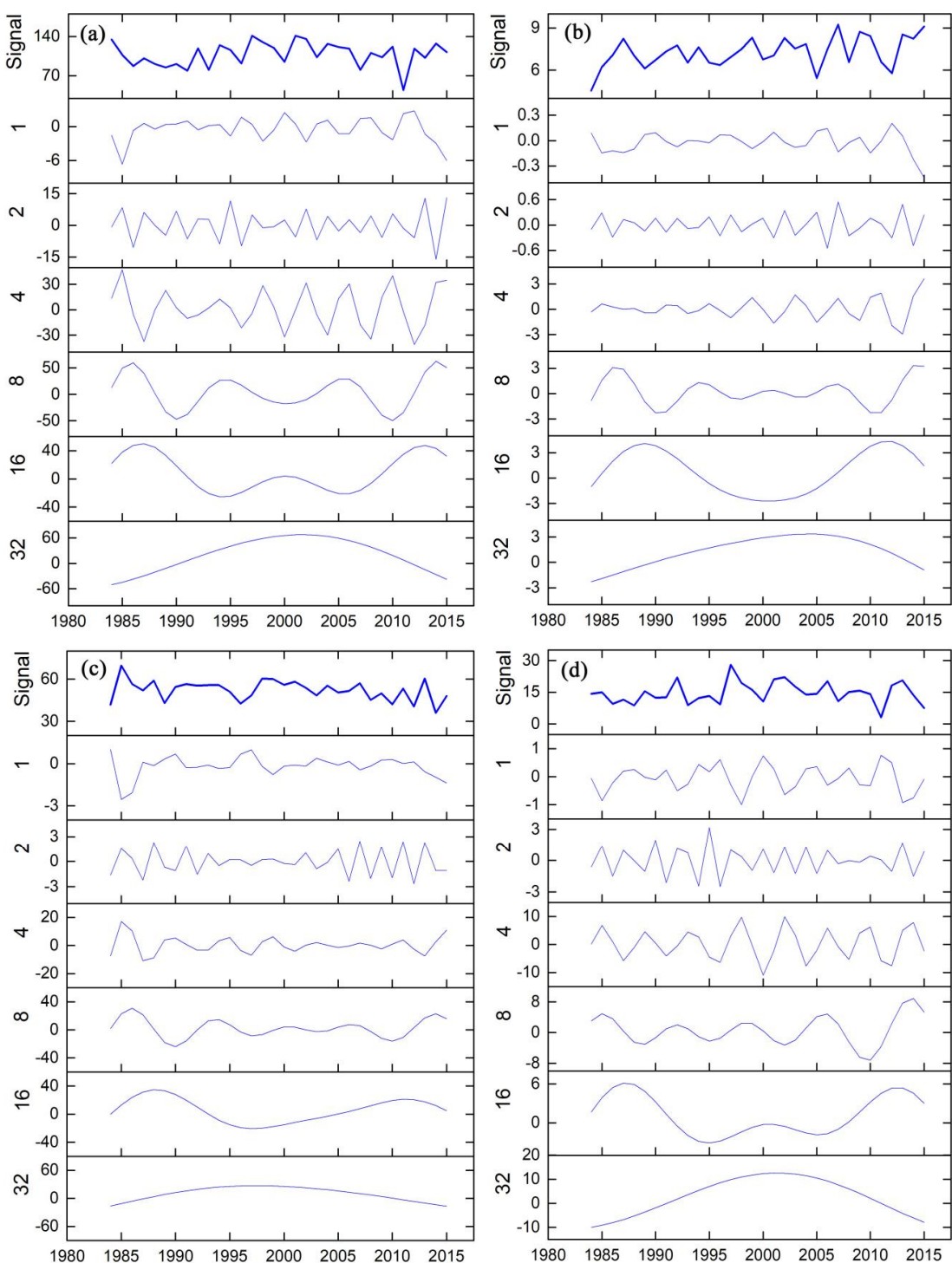

**Figure 4.** Multi-time-scale variations of rainfall (**a**), temperature (**b**), evaporation (**c**) and runoff (**d**) in spring.

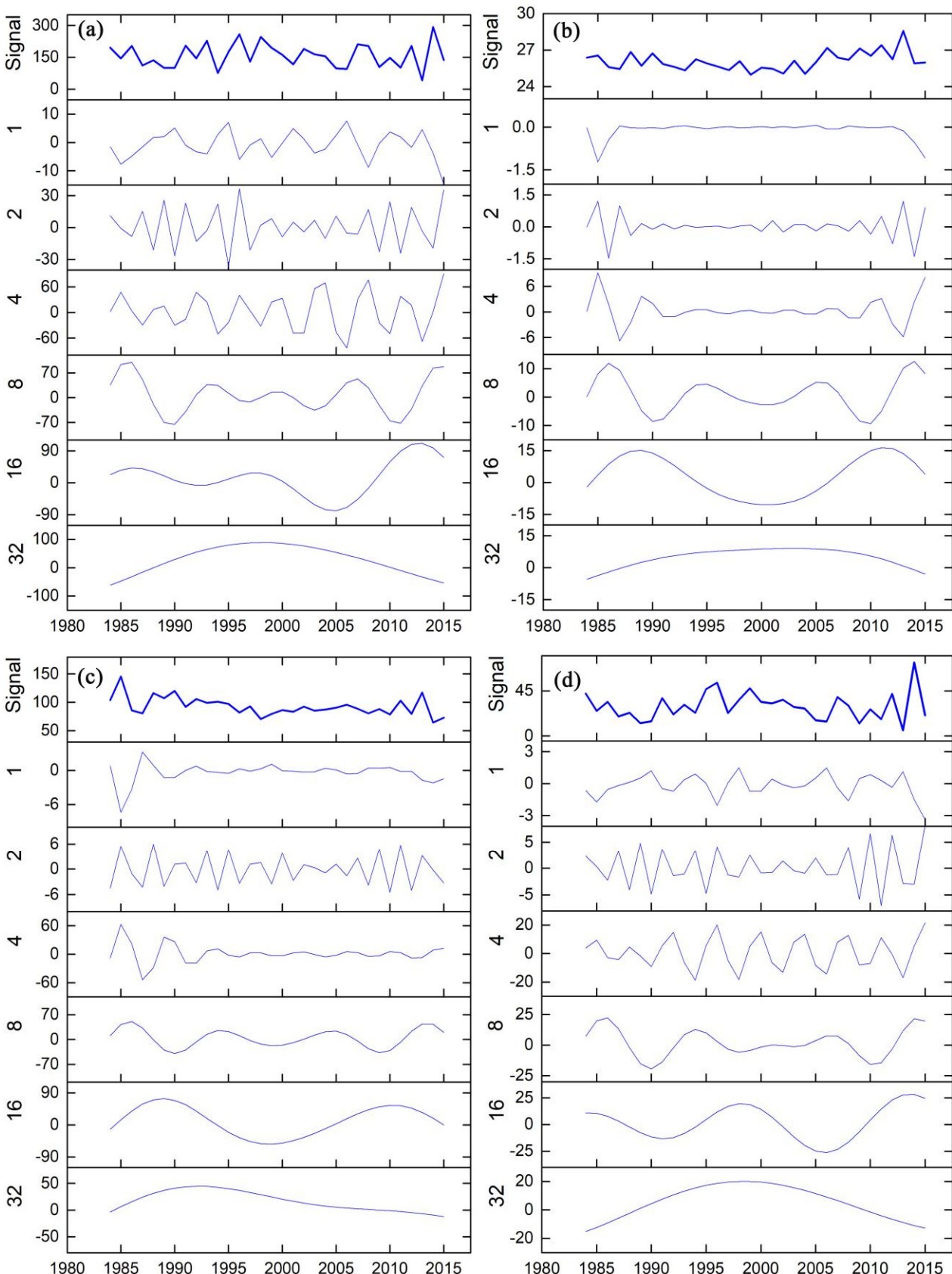

**Figure 5.** Multi-time-scale variations of rainfall (**a**), temperature (**b**), evaporation (**c**) and runoff (**d**) in summer.

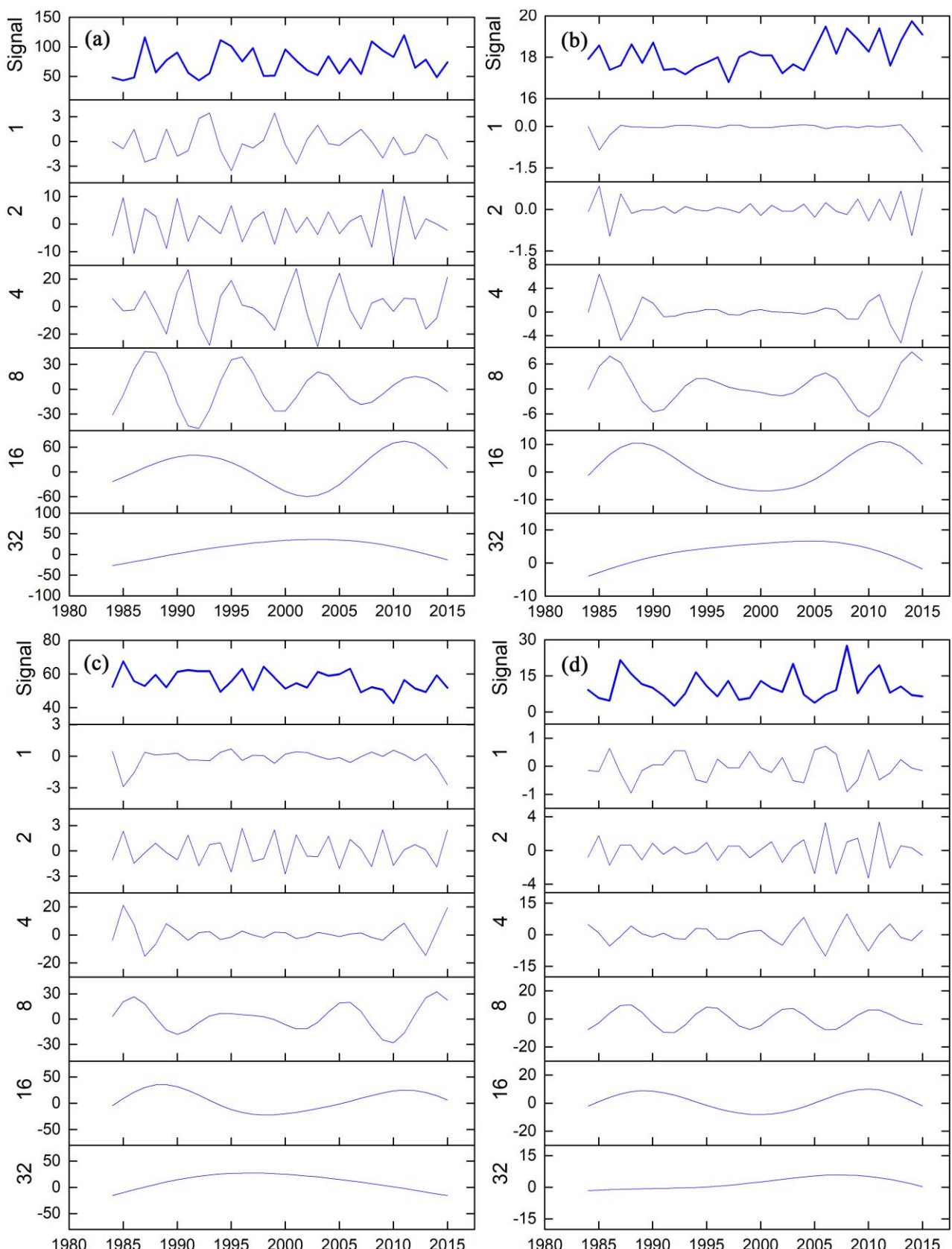

**Figure 6.** Multi-time-scale variations of rainfall (**a**), temperature (**b**), evaporation (**c**) and runoff (**d**) in autumn.

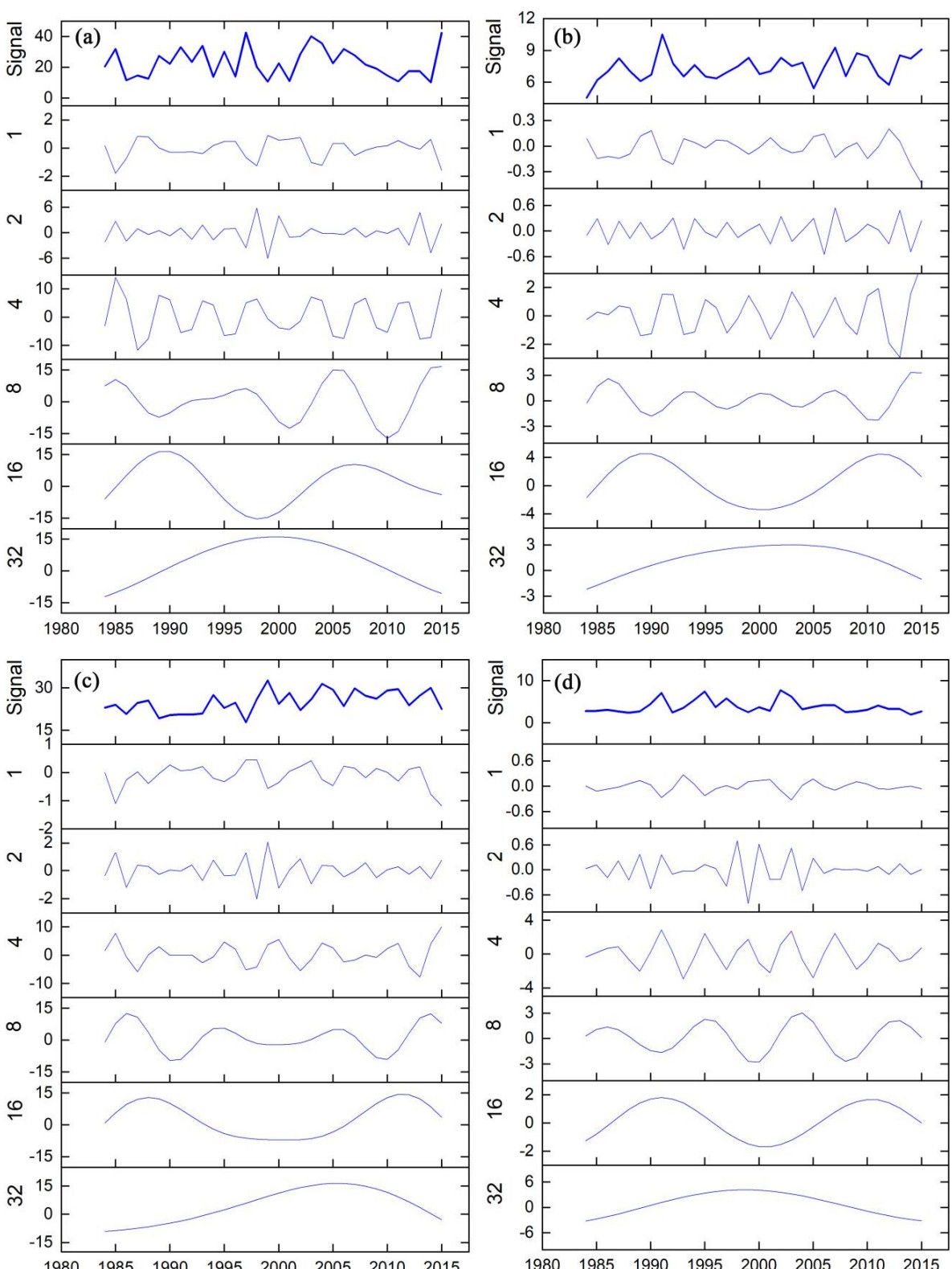

**Figure 7.** Multi-time-scale variations of rainfall (**a**), temperature (**b**), evaporation (**c**) and runoff (**d**) in winter.

5.1.3. Annual Variation

It has been found that rainfall and runoff show the same oscillation characteristics in different annual scales and the oscillation is obvious at 1–2- and 16–32-year time scales (Figure 8). Temperature and evaporation oscillate at 4–32-year sales. Runoff, rainfall,

temperature and evaporation have the same oscillation characteristics at 4-, 8- and 32-year time scales. Runoff is mainly affected by rainfall, whereas evaporation is mainly affected by temperature. As a whole, it has been found that there is a strong effect of rainfall over runoff but a lesser effect of temperature and evaporation over runoff.

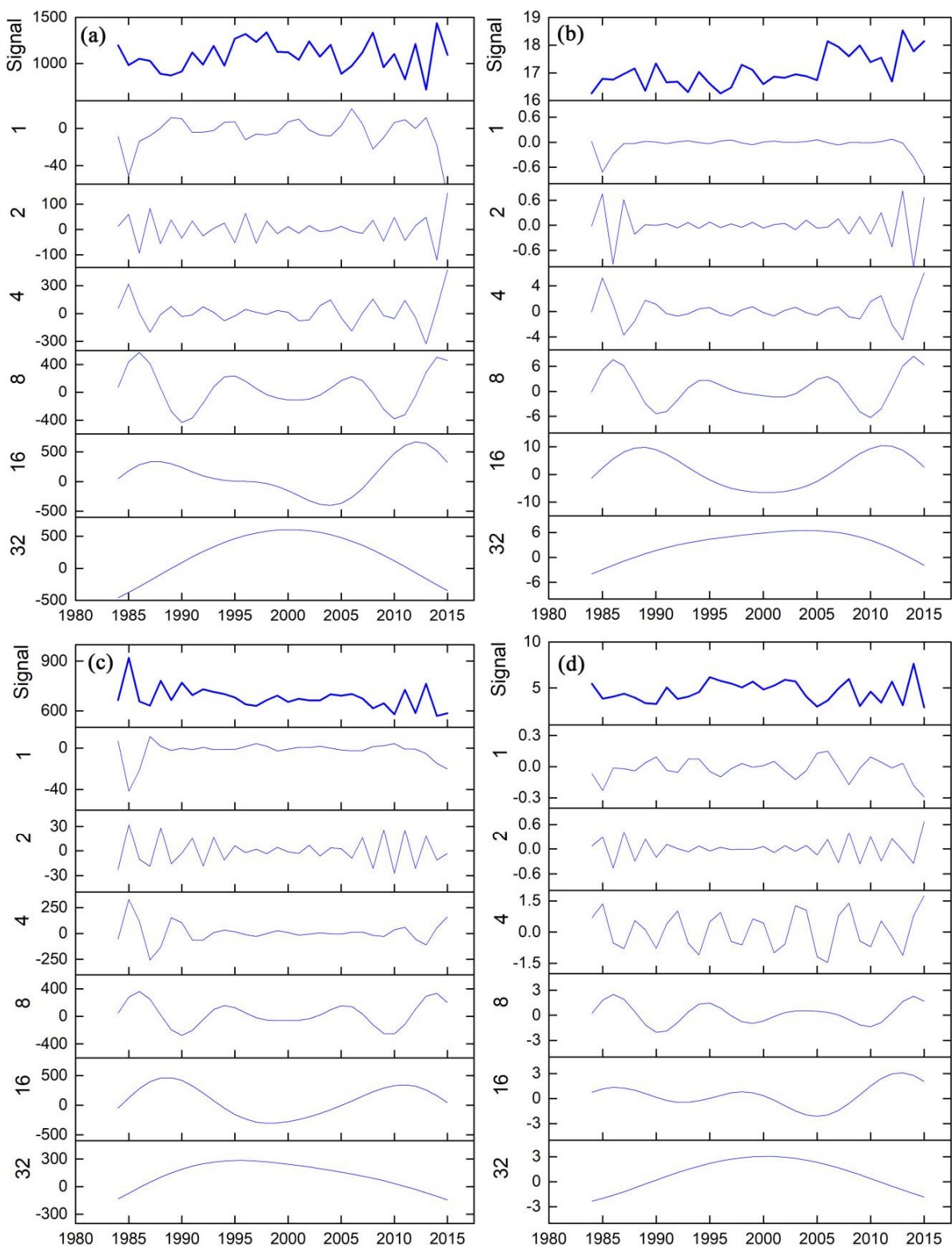

**Figure 8.** Multi-time-scale variations of annual rainfall (**a**), temperature (**b**), evaporation (**c**) and runoff (**d**).

### 5.2. Multi-Scale Evolution of Runoff and Climatic Factors Analyzed by CWT

As shown in Figure 9, the higher the spectral value (that is, the stronger the oscillation energy) is, the more significant the periodic oscillation passes the 0.05 confidence level. Except for the breakpoints at 8–16-month time scales in 1990 in Figure 9—Month Q, a main cycle always exists at 12-month time scales, which reflects the overall and significant periodic variation characteristics of monthly runoff. Several subcycles appear at approximately 36-month time scales (1990–2000) and 18–24-month time scales (1993–1997 and 2007–2012), which are related to the significant increase in rainfall in this period. The cycle at 4–6-month time scales (1984–2015) fluctuates in the time domain.

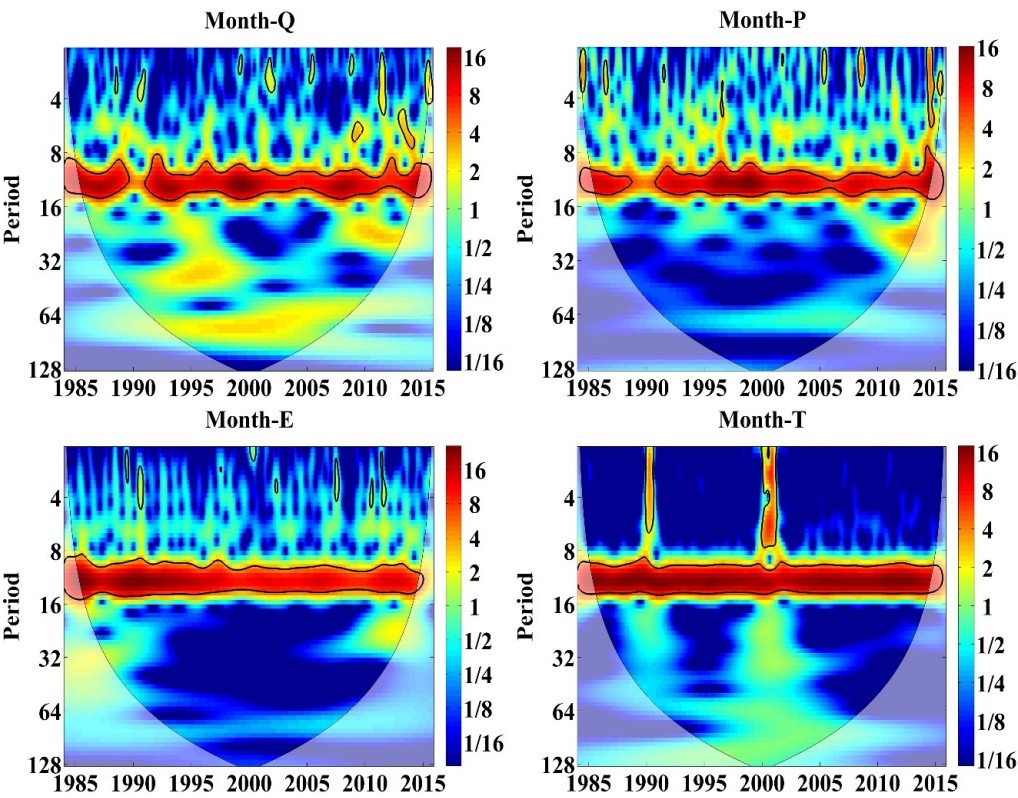

**Figure 9.** The continuous wavelet power spectra of monthly runoff (Month Q), rainfall (Month P), evaporation (Month E) and temperature (Month T) in the Yinjiang River watershed. The thick black contour designates the 5% significance level against red noise and the cone of influence (COI) where edge effects might distort the picture is shown as a lighter shade.

There are three subperiods with different significant levels of monthly rainfall in the time domain (Figure 9—Month P). The subperiod at 64-month time scales (1993–2008) indicates that the runoff has an important characteristic at 5-year time scales and a basically stable periodic variation. The subperiod at 24-month time scales (2007–2013) and at 18-month time scales (1993–1997) denotes that rainfall exerts a significant impact on runoff. Rainfall has similar fluctuation characteristics to runoff at 4–6 month time scales (1985–2015) and shows large fluctuations in the time domain.

There are three subcycles at 32-month time scales (1988–1992), 24-month time scales (2007–2013) and 6-month time scales (1984–2015) with different significant levels of monthly rainfall in the time domain (Figure 9—Month E). The fluctuation characteristics of evaporation are similar to those of runoff and rainfall at 6-month time scales. Obvious differences are also observed in the energy distribution characteristics in different periods, which indicates that rainfall and evaporation have obvious local characteristics of subperiodic variation consistent with those of runoff, but their energy is relatively weakened.

The monthly temperature is close to the monthly evaporation periodic bandwidth without interruption (Figure 9—Month T), which reflects the global and significant periodic variation characteristics of the monthly temperature and evaporation and indicates that the monthly temperature may affect the runoff mainly by changing the monthly evaporation. The monthly temperature has significant high-energy characteristics in the time domain around 1990 and 2000. The energy is strong at high frequencies below the scale of 1–8 months and weak at low frequencies after 8 months, but the periodic bandwidth increases with the scale.

It has also been found that a main period for runoff and climatic factors appears at 12-month time scales, which indicates that the periodic changes in hydrometeorology are mainly reflected in the annual scale. The discontinuous period and periodic bandwidth of climatic factors are basically consistent with the monthly runoff. The monthly runoff is consistent with the monthly rainfall, and the monthly temperature is consistent with the monthly evaporation. In addition, it has been found that the monthly runoff, rainfall and evaporation have the significant global fluctuation characteristics at high-frequency scales of below 8 months, whereas temperature exhibits only local fluctuation characteristics in 1990 and 2001 but significant impacts over 12-month time scales from 1995 to 2005. It can be concluded that rainfall is the main factor that affects runoff change in high-frequency regions and temperature and evaporation are the main factors in low-frequency regions.

As can be shown in Figures 10 and 11, runoff in spring has a main cycle at 4–6 years (1995–2000) and a subcycle at 1–2-year time scales (1990–1997). Runoff in summer is an insignificant period on the 4-year time scales (1990–2000). There are global insignificant characteristics for the main period of runoff in autumn at 1–2-year time scales (2006–2010) and the subperiod at 7-year time scales (1993–2005).

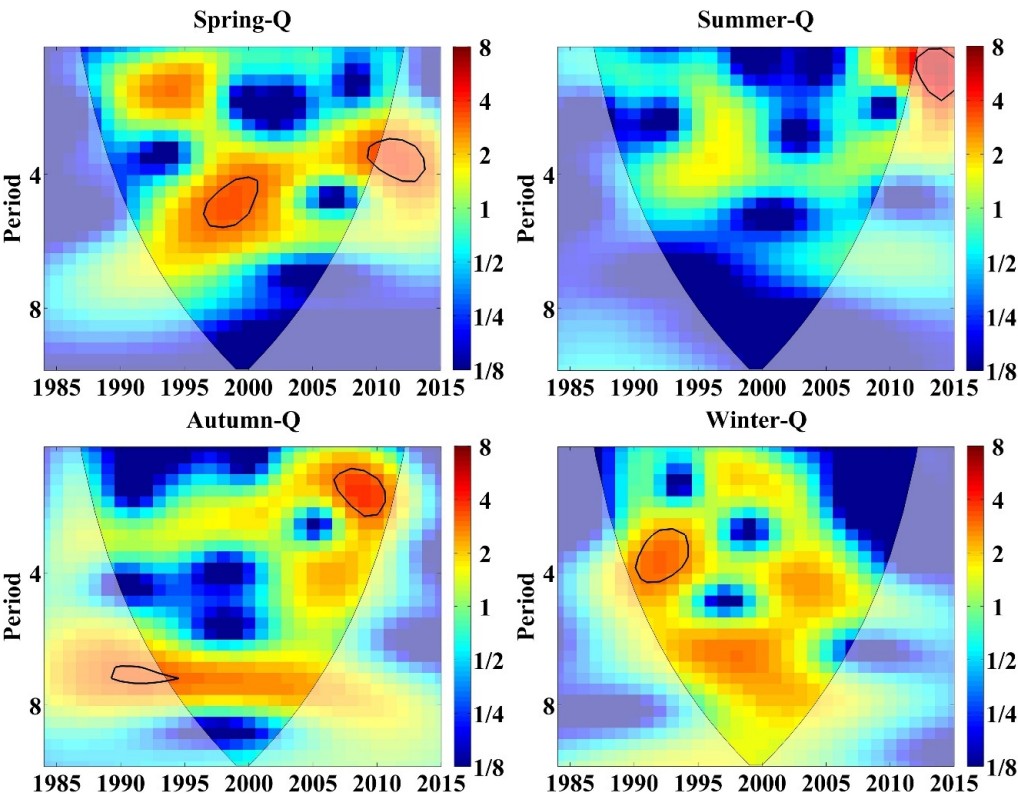

**Figure 10.** The continuous wavelet power spectra of seasonal runoff in the Yinjiang River watershed. The thick black contour designates the 5% significance level against red noise and the cone of influence (COI) where edge effects might distort the picture is shown as a lighter shade.

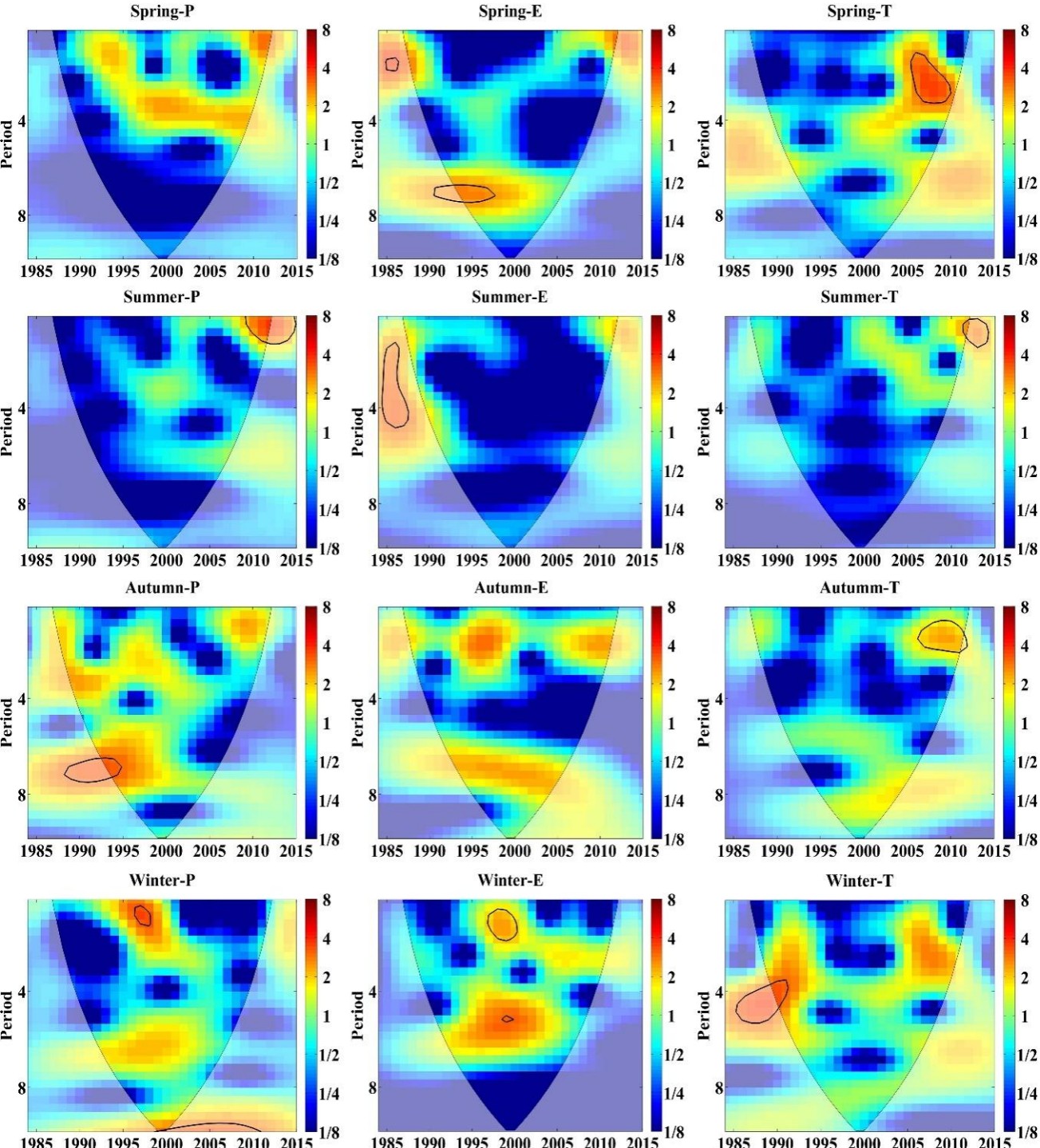

**Figure 11.** The continuous wavelet power spectra of seasonal rainfall, evaporation and temperature in Yinjiang River watershed. The thick black contour designates the 5% significance level against red noise and the cone of influence (COI) where edge effects might distort the picture is shown as a lighter shade.

The evolution characteristics of rainfall, temperature, evaporation and runoff vary greatly in spring, autumn and winter, but they are relatively stable in summer. They have great variations in high-frequency scales in spring and in low- and high-frequency scales in autumn, as well as periodicity in middle- and high-frequency scales in winter. Moreover, no

obvious periodicity is detected in summer, whereas the significant periodic characteristics exist in other seasons.

From the annual scale (Figure 12), annual runoff has a main period at 6-year time scales (1995–2007), and the main period of rainfall is insignificant on this scale in the same period. However, the high-power spectrum of rainfall at this scale shows that it has an important impact on runoff change. No significant main period of evaporation is observed in the entire valid spectrum period, but significant periodic variations in temperature occur at 1–6-year time scales in 1997–2003. The power spectrum value of annual temperature at 6–8-year time scales remains high but insignificant. Over the 8-year time scales, the periodicity is significant, but the period bandwidth is narrowed.

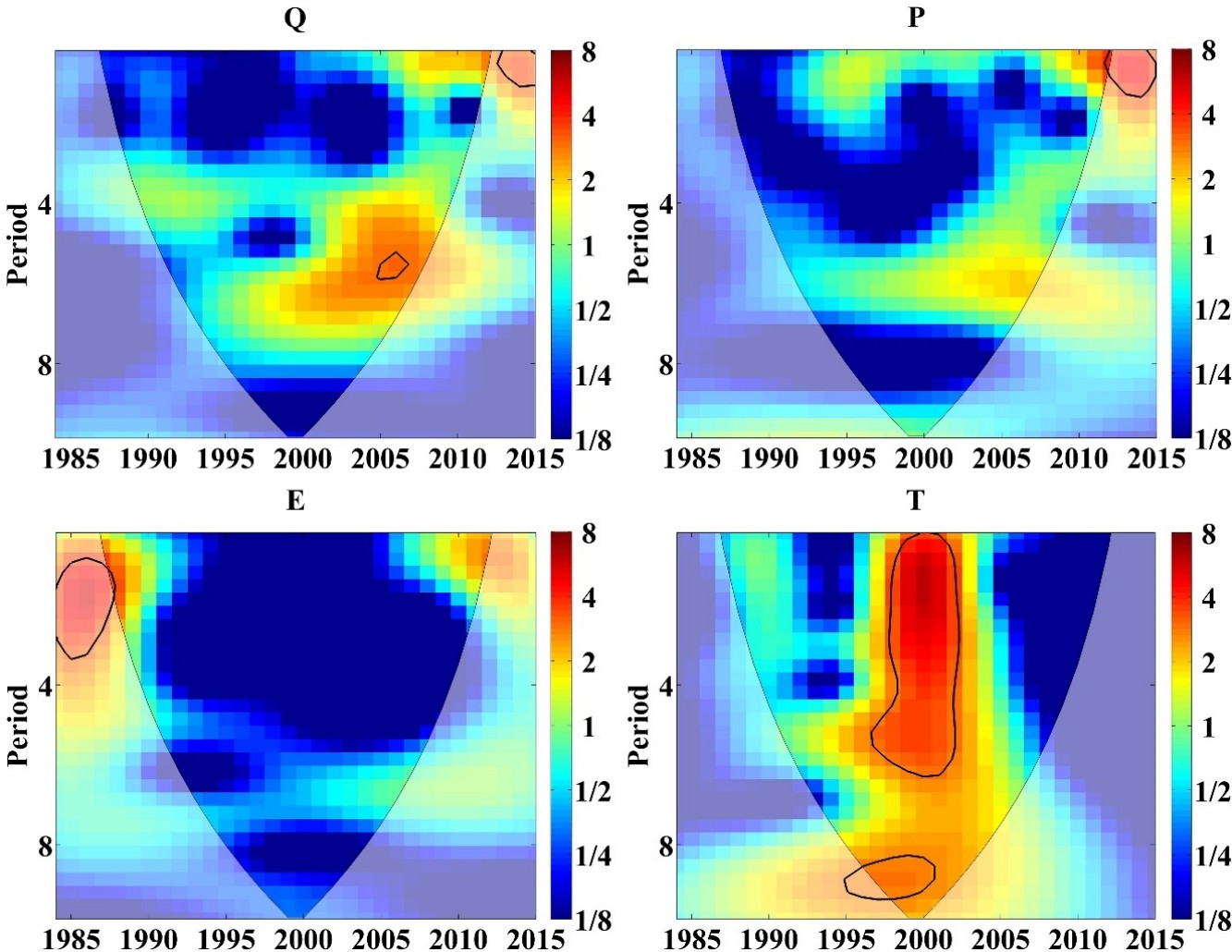

**Figure 12.** The continuous wavelet power spectra of annual rainfall, evaporation and temperature in Yinjiang River watershed. The thick black contour designates the 5% significance level against red noise and the cone of influence (COI) where edge effects might distort the picture is shown as a lighter shade.

*5.3. Response Characteristics of Runoff Changes to Climatic Factors at Different Time Scales*
5.3.1. Response of Runoff Changes to Climatic Factors on Monthly Scale

The cross wavelet power spectra of monthly rainfall and runoff, as shown in Figure 13, illustrate that the interaction between monthly rainfall and runoff is mainly concentrated in the main cycle at 12-month time scales from 1984 to 2015, which indicates a significant correlation between them at 1-year scale. The interaction between monthly rainfall and runoff is also shown in two subcycles at approximately 24-month time scales (1993–1996

and 2007–2013) and 72-month time scales (1993–2008). The energy difference in main cycles and subcycles in the time domain reflects that rainfall in different years has different regulating effects on runoff change.

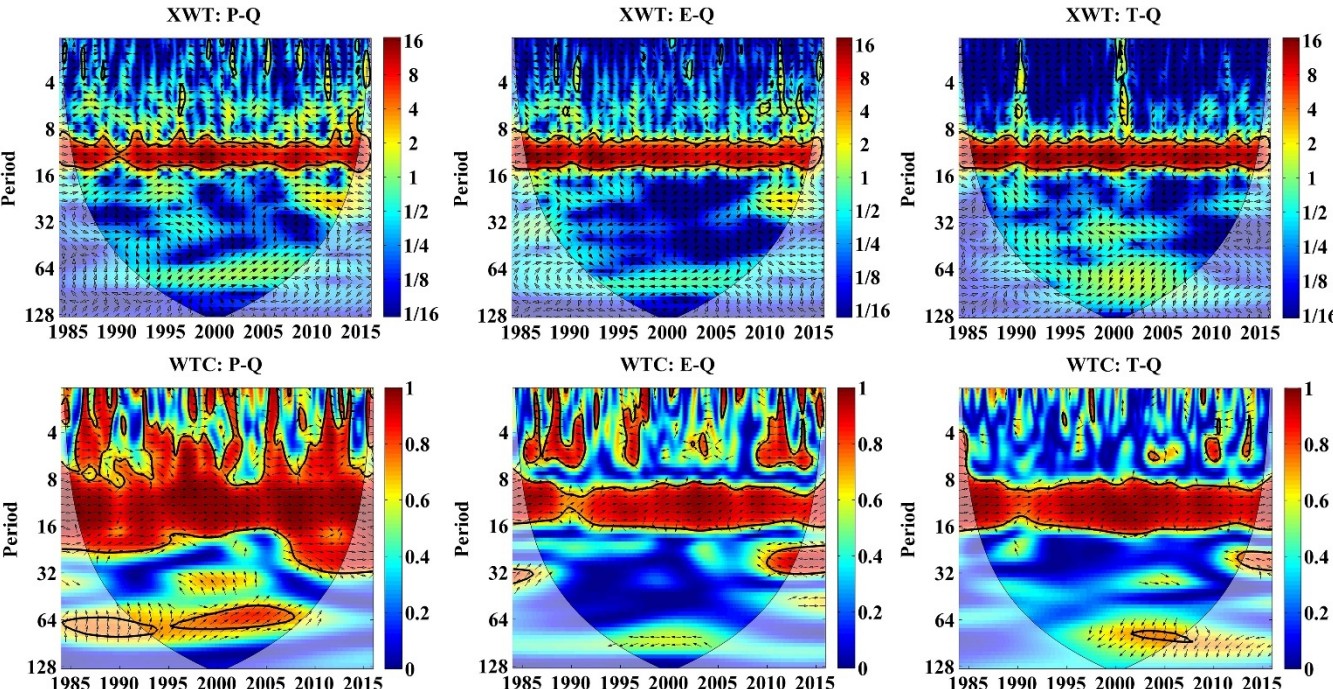

**Figure 13.** Cross wavelet spectra and wavelet coherence spectra of monthly rainfall, temperature and evaporation with runoff in the Yinjiang River watershed from 1984 to 2015, respectively. The thick black contour designates the 5% significance level against red noise and the cone of influence (COI) where edge effects might distort the picture is shown as a lighter shade. Arrows denote relative phase difference: The arrows from left to right indicate that the influencing factors and runoff are in the same phase, which implies a positive correlation; the arrows from right to left indicate an inverse phase, which implies a negative correlation; the downward arrows indicate that the influence factor is 90° ahead of the runoff change and the upward arrows indicate that the influence factor is 90° lagging the runoff.

The cross wavelet coherence spectra can compensate for the lack of correlation analysis of the cross wavelet power spectra in the low-energy region. The cross wavelet condensation spectra has a larger time–frequency domain space compared with the cross wavelet power spectra. In addition to the positive correlation at 12-month time scales between 1984 and 2015, a positive correlation is also observed in subperiods between 1991 and 2008 (72-month time scales) and 1984 and 2015 (4–8 months). By contrast, a negative secondary cycle occurs in 1996–2004 (36-month time scales). The contribution of rainfall to runoff in the Yinjiang River watershed can also be determined from the phase relationship between the cross wavelet power spectra and the cross wavelet condensation spectra. The main periods of 4–8-, 12- and 72-month time scales show a significantly positive correlation. This result is mainly because soil moisture is easy to be saturated due to the increase in rainfall on this scale, thus accelerating the formation of slope runoff within the river watershed and forming an effective replenishment for the river. On the contrary, the phase is negatively correlated in the subcycle at 36-month time scales, which indicates that the influence of rainfall fluctuation on runoff changes is negative. On the one hand, runoff will increase with the increase in rainfall on the river recharge. On the other hand, the increasing rainfall will increase vegetation coverage, enhance water conservation, increase evaporation and eventually reduce river recharge. The high-correlation region of the cross wavelet conden­sation spectra is basically consistent with the high-energy region of the cross wavelet power

spectra. A strong correlation exists between the monthly rainfall and runoff at 64-month time scales from 1995 to 2008, and the phase angle is 30°, which indicates that the rainfall lags behind the runoff for 1 month.

The monthly evaporation, temperature and runoff have main periods of 12-month time scales in high-energy areas. In low-energy regions, not only a high correlation at 12-month time scales but also a local significant correlation between the effects of temperature and evaporation on runoff changes at 1–8-month time scales are determined. Thus, temperature is mainly regulated indirectly by controlling evaporation. The influence of evaporation on runoff will superimpose the influence of temperature due to the indirect regulation of temperature on runoff because their influence on runoff is generally consistent in different scales but slightly different in different years.

The effects of monthly rainfall, temperature and evaporation on runoff are positively correlated in the main cycle at 12-month time scales, which indicates that their effects on runoff are positive and mainly at the annual scale. The main periodic bandwidth of rainfall on runoff changes is wider than that of temperature and evaporation, which indicates that rainfall is the main factor that affects runoff variation. In other cycles, the phases of rainfall's impacts on runoff changes are the interlacing phenomena of positive and negative, whereas the phases of temperature and evaporation that affect runoff changes are mainly negative. Accordingly, the impact of rainfall fluctuation on runoff changes on this scale is both positive and negative, whereas that of evaporation is always negative. However, temperature and evaporation have negative effects on runoff in each subcycle, which may be because evaporation increases with the increase in temperature, thus reducing runoff recharge.

The response of runoff to rainfall is timely in the high-energy region and the low-energy significant-correlation region. There has been shown a positive correlation with a smaller phase angle, but there is also a slight lag at 16-month time scales. The phase angles of evaporation, temperature and runoff range from 30° to 45°, which demonstrates that runoff changes have lagged behind temperature and evaporation for 1–2 months. The similarities of the effects of monthly temperature and evaporation on runoff changes have also proved that temperature indirectly affects runoff changes by changing evaporation.

5.3.2. Response of Runoff Changes to Climatic Factors on Seasonal Scale

The effect of rainfall on runoff changes in the four seasons has a significant high-energy region from the XWT power spectra (Figure 14). There are higher power spectrum values, stronger influences and wider time domains in spring and autumn. In spring, the related regions are mainly in the main cycle at 4-year time scales from 1998 to 2012 and the subcycle at 1-year scale from 1991 to 1995. In autumn, the relevant regions are mainly in the main cycle at 7-year time scales from 1992 to 2000 and the subcycle at 2-year time scales from 1995 to 2000 and then are invisible.

The effect of evaporation on runoff changes presents some significant high-energy zones in four seasons. Energy is strongest in autumn and weakest in summer. The evaporation effect is significant at 6–8-year time scales from 1993 to 2005. In spring, summer and autumn, the locations of significant high-energy zones on the same scale are basically the same. The effect of evaporation on runoff in spring and summer differs on different time scales. In autumn and winter, the energy distribution of the effect of evaporation on runoff changes is similar to that of rainfall on runoff changes, but the phase relation is opposite. Accordingly, the regulation of rainfall to runoff changes is positive, whereas that of evaporation is negative. This phenomenon may be caused by drought, less rain and strong evaporation, which can directly reduce runoff.

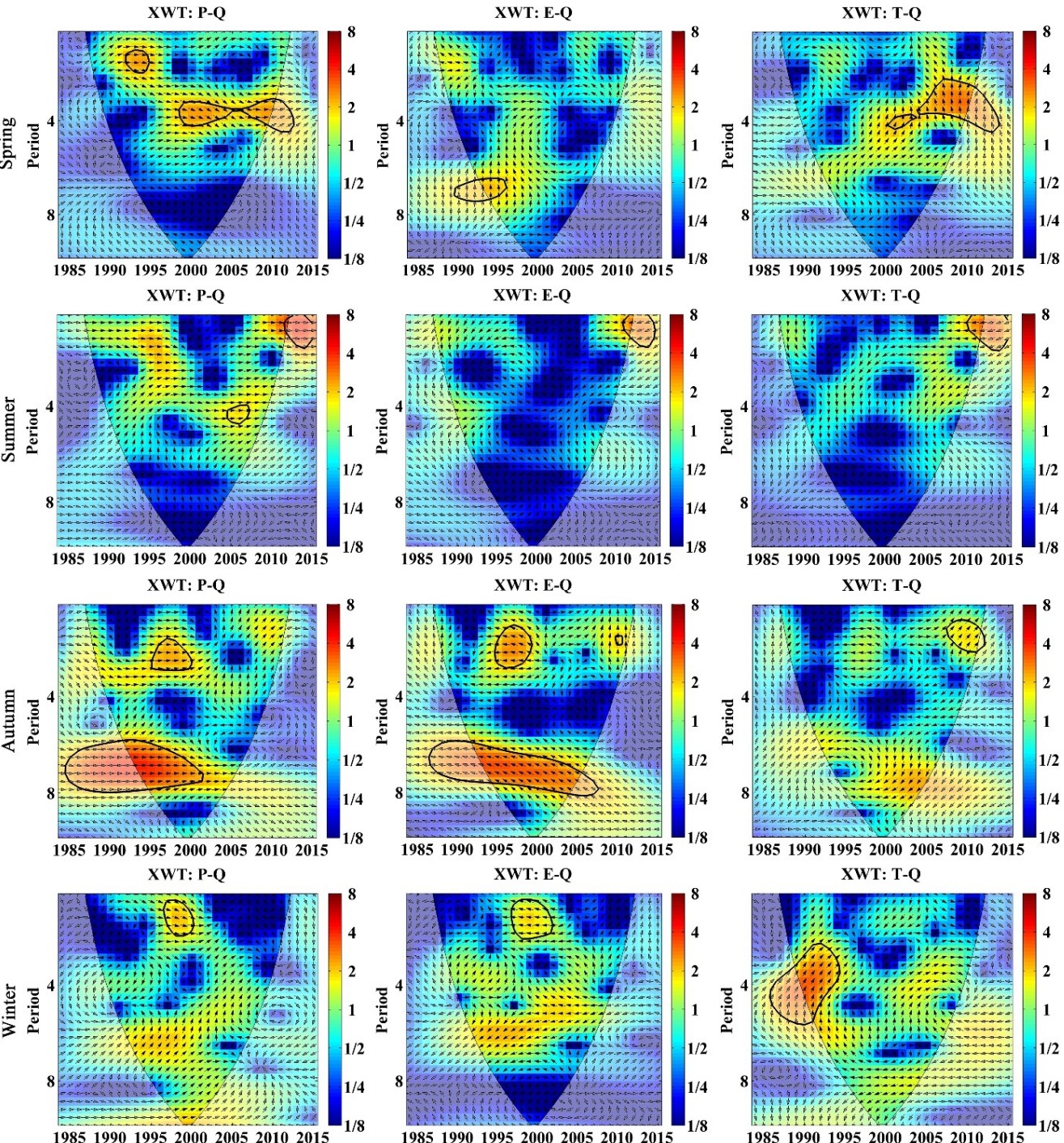

**Figure 14.** XWT between runoff and rainfall, evaporation and temperature on monthly scale. The thick black contour designates the 5% significance level against red noise and the cone of influence (COI) where edge effects might distort the picture is shown as a lighter shade. Arrows denote relative phase difference: The arrows from left to right indicate that the influencing factors and runoff are in the same phase, which implies a positive correlation; the arrows from right to left indicate an inverse phase, which implies a negative correlation; the downward arrows indicate that the influence factor is 90° ahead of the runoff change and the upward arrows indicate that the influence factor is 90° lagging the runoff.

The effect of temperature on runoff also has significant high-energy regions in the four seasons. The power spectrum value is high, and the influence is strong in spring and winter. The influence of temperature in spring mainly concentrates on the main cycle at 4-year time scales from 2000 to 2014, and it is consistent with the influences of rainfall and evaporation on runoff at more than 6-year time scales. The effect of temperature on runoff changes mainly concentrates on the main cycles at 1-year time scale in summer and at 2-year time scales in autumn from 2007 to 2010. However, the effect of temperature on runoff in autumn has a subcycle at approximately 8-year time scales with high energy from 1995 to 2005, and it is similar to that of rainfall and evaporation at the same scale in the same time domain. The changes in temperature and evaporation in summer are ahead of runoff change, but the lead time of temperature ahead of runoff is greater than that of evaporation in summer. In addition, the effect of temperature on runoff changes in winter has a significant high-energy region at 4-year scale from 1990 to 1995, and that of rainfall and evaporation on runoff is stronger in the same time–frequency domain. Seasonally, the influence of temperature on runoff is similar to that of rainfall on runoff in the energy distribution in spring, which indicates that the increase in temperature results in increased rainfall and thus increases rainfall supply to runoff. In autumn, the influences of temperature, evaporation and rainfall on runoff have a consistent feature in energy distribution, which also shows that temperature has an important impact on evaporation and rainfall and leads to the same effect on runoff.

From the WTC condensation spectra of rainfall, temperature and evaporation with runoff in the four seasons (Figure 15), the highly correlated area of rainfall impacts runoff changes with an increase in years and scales, and it changes from 1-year scale in 1990–2000 to 4-year scale in 1995–2005. The main period of the impact of rainfall on runoff in summer is concentrated on the high-frequency scales, and the bandwidth tends to widen, which indicates that the period tends to be stable. The effect of rainfall on runoff in autumn is mainly manifested in the main period at approximately 8-year time scales from 1992 to 2005, with a wide bandwidth and an extremely stable period. The significant-correlation area of rainfall on runoff in winter is concentrated at 1-year and 7-year time scales from 1993 to 2003, and the influence is relatively weak. The significant-correlation region of evaporation in each season is consistent with the high-energy region of the XWT power spectrum, but it is more significant in autumn over 4-year time scales. Although the bandwidth in autumn is narrowed from 1990 to 2006, it still has a wide periodic bandwidth, which is similar to the impact of rainfall on runoff in the same season, indicating that the runoff change is mainly affected by rainfall and evaporation in autumn. According to phase characteristics, the phase relation between rainfall and runoff is positive, which implies that the influence is always positive, whereas the potential correlation between evaporation and runoff is negative; hence, the influence is negative, and their impact on runoff has a common main period at 8-year time scales. The influence of temperature in spring, summer and winter on runoff is relatively significant, and it is concentrated at the 4-year scale from 2000 to 2010 in spring, at the 1–4-year time scales from 2004 to 2010 in summer and at the 4-year time scales from 1990 to 2000 in winter. This finding indicates that the influence of temperature on runoff has significant differences in scale and time domain in different seasons, that is, it has significant local characteristics rather than global characteristics. As a whole, the main periodic bandwidth of the significant-correlation region in each season tends to widen, and the time-domain range of the correlation region also increases. The frequency-domain structure of the significant-correlation region of the WTC is basically consistent with that of the XWT high-energy region.

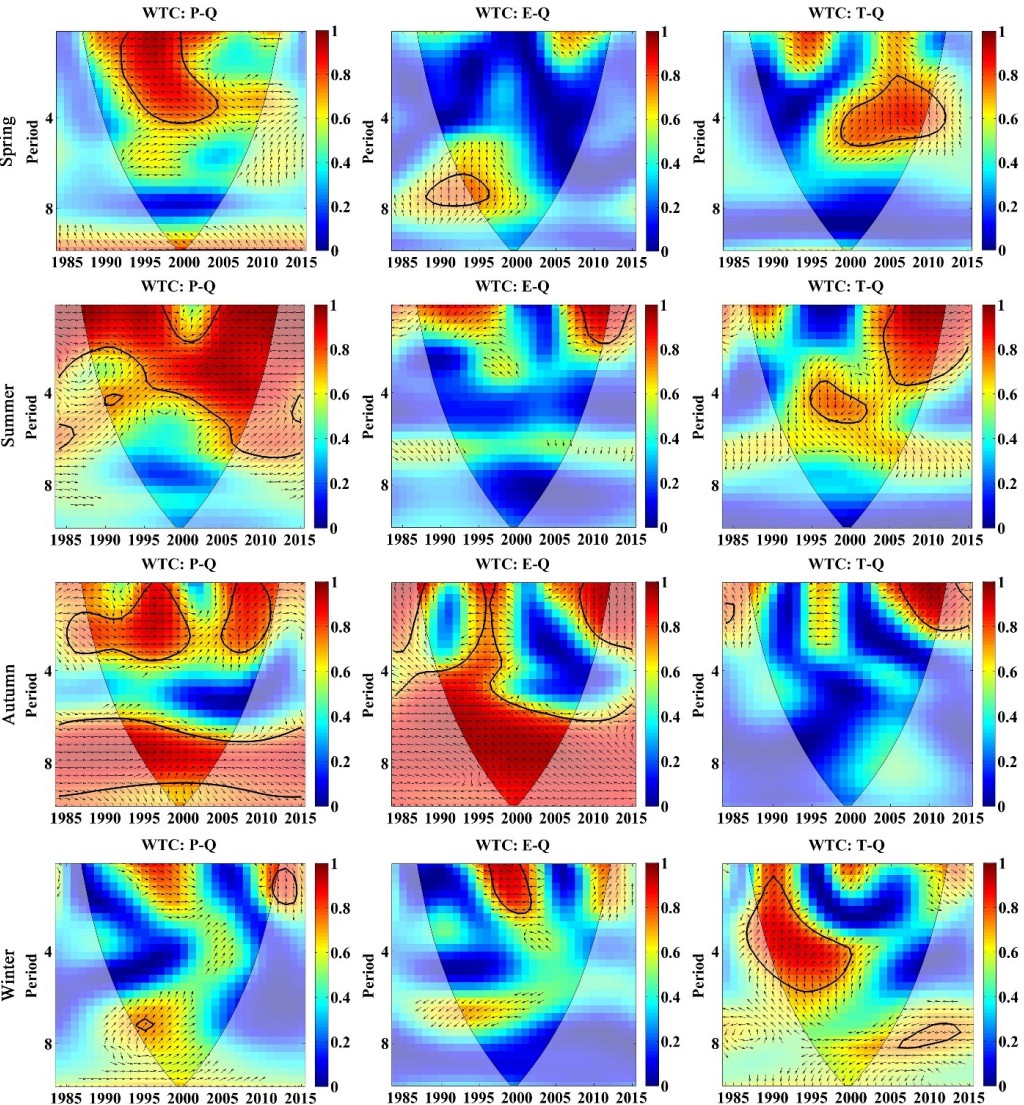

**Figure 15.** WTC between runoff and rainfall, evaporation and temperature on monthly scale. The thick black contour designates the 5% significance level against red noise and the cone of influence (COI) where edge effects might distort the picture is shown as a lighter shade. Arrows denote relative phase difference: The arrows from left to right indicate that the influencing factors and runoff are in the same phase, which implies a positive correlation; the arrows from right to left indicate an inverse phase, which implies a negative correlation; the downward arrows indicate that the influence factor is 90° ahead of the runoff change and the upward arrows indicate that the influence factor is 90° lagging the runoff.

In summary, runoff changes are mainly affected by rainfall and temperature in spring, mainly by direct rainfall recharge. That temperature increases rainfall and evaporation is the reason why its phase relation presents a positive and negative interlacing phenomenon in spring. In summer, runoff is mainly affected by direct rainfall recharge, the effect of evaporation on runoff changes is negative, and the positive effect is mainly reflected at more than 5-year time scales. In autumn, runoff change is affected by a small amount of rainfall supply and runoff loss is caused by evaporation. In winter, runoff is mainly affected by temperature because the rainfall in karst areas cannot form the effective recharge for runoff due to the drought and minimal rain; however, the temperature can indirectly adjust runoff changes by changing evaporation.

### 5.3.3. Response of Runoff Changes to Climatic Factors on Annual Scale

For XWT P–Q (Figure 16), a strong-influence period occurs in 2005, which indicates that the intensity of interaction between rainfall and runoff changes after the sudden variation in runoff in 2003. This phenomenon also implies that climatic factor is the main driving factor for the recent runoff increase. The high-intensity effects of P–Q present a significantly positive correlation and occur mainly at approximately 6-year time scales in the period from 2000 to 2010. The effect of P–Q passes the test of the red noise standard spectrum at the 0.05 significance level with a phase angle of 60°, which indicates that runoff is ahead of rainfall by 2 years on the 6-year time scales. The highly significant correlation after 2010 occurs on the 1–2-year time scales with consistent characteristics.

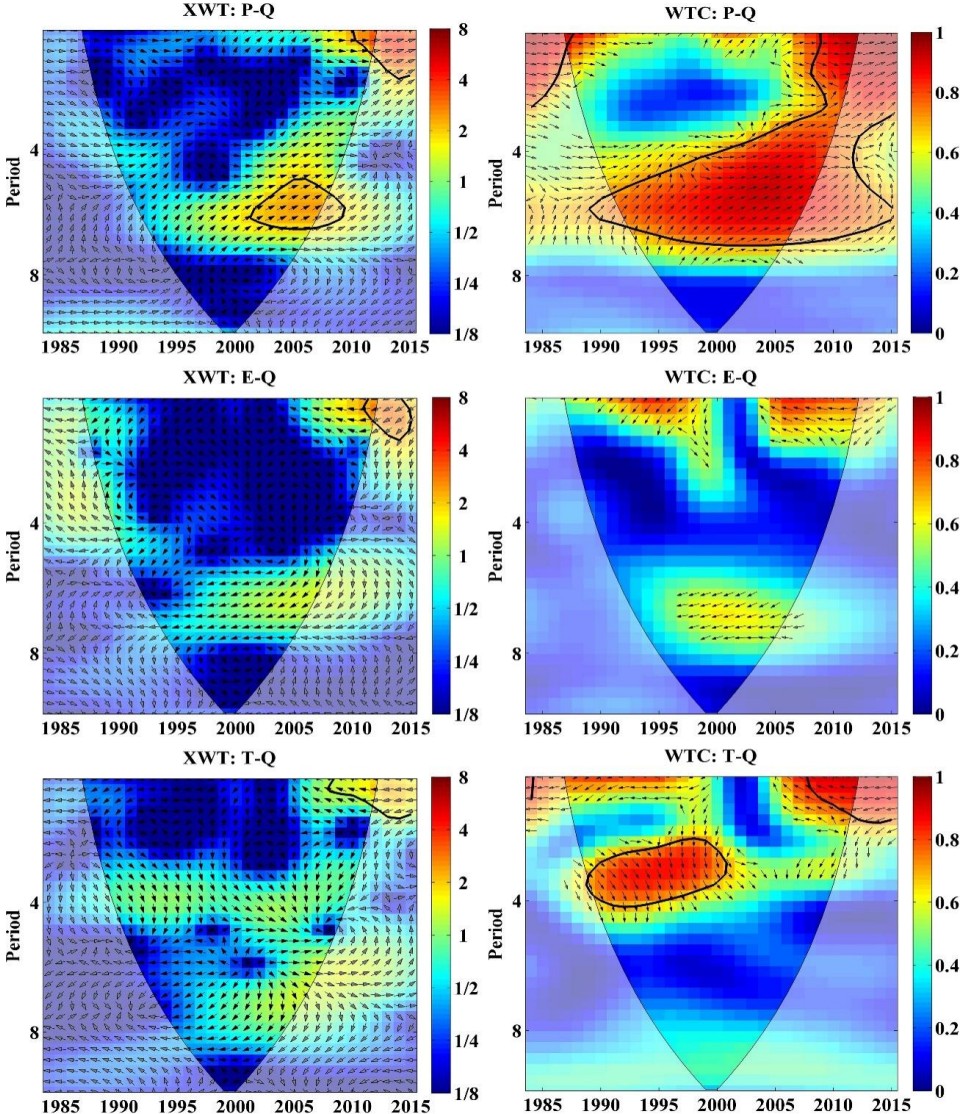

**Figure 16.** The XWT and WTC for annual rainfall (P) and runoff (Q), evaporation (E) and runoff (Q) and temperature (T) and runoff (Q) in the Yinjiang River watershed from 1984 to 2015. The thick black contour designates the 5% significance level against red noise and the cone of influence (COI) where edge effects might distort the picture is shown as a lighter shade. Arrows denote relative phase difference: The arrows from left to right indicate that the influencing factors and runoff are in the same phase, which implies a positive correlation; the arrows from right to left indicate an inverse phase, which implies a negative correlation; the downward arrows indicate that the influence factor is 90° ahead of the runoff change and the upward arrows indicate that the influence factor is 90° lagging the runoff.

For WTC P–Q, a significant high-correlation region on 4–7-year time scales exists during the entire period, which indicates that runoff is strongly affected by rainfall. From the phase diagram of P–Q, runoff has shown 2–3-year time scales ahead of rainfall in 1990–2000. Thus, the change in runoff is mainly affected by human activities. After the 2000s, the phase angle decreases gradually, indicating that runoff is gradually aggravated by rainfall. According to the results of the XWT and WTC of P–Q, the high-energy region and high-correlation region of P–Q are mainly concentrated around the middle of 2010, and the main cycle is mainly at 6-year time scales.

For XWT E–Q, since 2010, it has a highly significant correlation on the 1–2-year time scales, which demonstrates that evaporation has a significant impact on runoff at this time scale. For WTC E–Q, the E–Q cycle mainly concentrates at 1–2-year time scales during the period from 1990 to 2000 and from 2005 to 2015. The E–Q cycle mainly concentrates on 6–8-year time scales from 1995 to 2005, which indicates that the effect of evaporation on runoff is small. The E–Q phase correlation also shows that the E–Q phase correlation is an inverse phase with a phase angle of 45° from 1990 to the end of 2010 at 1–2-year time scales, which implies that evaporation is 1–2 years ahead of runoff. After 2005, evaporation remains in an inverse phase with runoff, with an initial phase angle of approximately 30 (at 1-year time scale), and then decreases and finally increases. The relationship between evaporation and runoff changes from lag to consistency to advance, which indicates that evaporation pays an important role on runoff changes. The results of XWT and WTC show that the phase angle on 7-year time scales is approximately 45° from 1995 to 2005, indicating that E is approximately 1.5 years ahead of Q.

For XWT T–Q, there is a high-energy region existing at 1–2-year time scales after 2005, which has passed the test of the standard spectrum of red noise at the 0.05 significant level. Therefore, the influence of temperature on runoff suddenly strengthens around 2005. However, the influence is relatively weak on the 4-year time scale and 6–8-year time scales, and it has not passed the test of red noise standard spectrum at the 0.05 significance level. The WTC results show two low-energy areas with the greatest impact. One is at the time of 3–4-year time scales in the period from 1990 to 2002, in phase with the phase angle between 60° and 70°, showing that temperature is ahead of runoff for more than 2 years. The other is at 1–2 year time scales after 2005, in which T–Q shows a negative correlation in the opposite phase with the initial phase angle of 45° and then gradually reduces to 0°. The above results show that the effect of temperature on runoff gradually changes from lag to consistency, indicating that the effect of temperature on runoff changes is increasingly obvious.

Overall, the interaction of rainfall with runoff changes at 6-year time scales across the entire period. However, the effects of temperature and evaporation on runoff changes are locally significant. The effect of evaporation on runoff changes is similar to that of temperature and has obvious local characteristics, mainly on small cycles.

## 6. Discussion

### 6.1. Multi-Scale Effects of Rainfall on Runoff Changes

Although the interaction between rainfall and runoff is positively correlated on the whole, the temporal effects are inconsistent in different time domains and scales. The effect of rainfall on runoff is ahead, lagging and consistent in time, ahead in high-frequency and low-frequency scales, lagging at medium-frequency scales (approximately 4 years) and consistent at 1-year scale and significant main periodic scales. For the leading effect, runoff may be mainly affected by early rainfall, which mainly occurs in rainy weather in spring and autumn. The surface and underground areas of karst are filled with soluble rocks with the main type of carbonate, which is vulnerable to erodible water that contains $CO_2$, thereby forming a large number of karst pipelines and fissures [51–53] over a long period of time and two sets of surface and underground hydrological systems [46]. Rainfall requires first to saturate soil water due to low soil moisture in karst areas [54–56]. Surface runoff is difficult to form with small rainfall due to the fragmented surface, steep and rugged

slopes, low runoff coefficients in slope surfaces and small river network density [57–61]. Rainfall is the main factor for runoff formation, and its intensity, duration and areas have great influences on runoff change. When raining heavier, rainwater may hardly infiltrate and leak and thus then increase runoff. If the rainfall intensity is smaller, most of the rainwater infiltrates into the soil and leaks through enormous karst fissure pipelines, which can reduce the runoff. The longer the duration of rainfall and the larger the area of rainfall, the easier the soil moisture will be saturated, and the runoff generated on the slope will inevitably be larger. Runoff monitoring studies on karst slopes show that light rainfall intensities (15~30 mm/h) generate subsurface lateral flow and underground fissure flow, whereas great rainfall intensities generate surface runoff in addition to subsurface and underground flows [62]. However, only a single rainfall of more than 60 mm on a karst slope can produce stable runoff because once the atmosphere rains [63], it immediately runs off into the ground through a broken surface and underground fissure, with distribution ratios of 27.8–78.0%, dominating the total flow yield. Therefore, the loss of rainfall and the formation of runoff in the slope surfaces of karst areas are much more difficult than those in non-karst areas. Only the last rainfall may form slope surface runoff under repeated rainfall because of the recharge of soil moisture first and the loss through fragmented surface leakage. Pre-rainfall mainly supplements soil moisture or leaks down through the broken surface to the pipeline and fissure. For all that, it has been found that all climatic factors exhibit a main cycle at 12-month time scales with runoff changes, which may show that the hydrometeorological processes in karst watersheds represent the same characteristics as those in non-karst watersheds at 1-year time scale (12-month time scale) periodic variations. This may be mainly because the impact of karst characteristics on hydrometeorological processes is mainly manifested on the slope scale, and all flows in the watershed will eventually converge to the outlet of watershed [45,46], which leads to the same annual periodic characteristics as those of non-karst watersheds. If the interval of multiple rainfall is long and the cumulative rainfall is less than 60 mm, it may lead to rainfall changes ahead of runoff on a monthly scale. If the cumulative rainfall is large under the condition of multiple short-term rainfall, the last few small rainfall events will produce obvious runoff on the slope after the saturation of soil water, and then the rainfall before the saturation of soil water will produce a leading effect on runoff. The effect of rainfall after soil water saturation on runoff changes will be synchronous because surface runoff would only occur when both soil and carbonate fissures and fractures are fully saturated with water [36,39]. Most of the rainfall is transported to the groundwater system through carbonate fractures and fractures, while the rainfall that can form surface runoff is very small [53,64].

In addtion, some studies have shown that antecedent rainfall and rainfall intensity are the major factors that control rainfall–runoff and soil erosion processes [65]. Rainfall intensity, slope angle and groundwater porosity [57] are the influencing factors of runoff changes mainly because the runoff mechanism caused by rainfall is different in years with different soil water contents. Different soil moisture contents are present in the early stage and the runoff generated by rainfall is also different in the year of rainfall approaching due to the different soil moisture contents in the early stage. In this case, the annual runoff depth is related to the rainfall year. For some places where recharging soil moisture by rainfall is difficult, the annual runoff depth is even related to the last rainfall year or even the previous years.

The process from rainfall to runoff will undergo seepage storage, slope overflow and channel flow collection. In karst watersheds, each process will be accompanied by underground leakage and the broken surface will affect the time, which greatly lengthens the lag time of runoff change. The influence of the changes in underlying surface conditions on runoff is a gradual process, but the influence of human activities on runoff is a catastrophic process. Therefore, the main reason why rainfall lags behind runoff is that human activities lead to catastrophic changes in runoff, especially land use changes, which destroy the original runoff production and confluence conditions. Such human activities as pumping and storing or introducing water into farmland can also lead to catastrophic changes in

runoff. Therefore, the effect of rainfall on runoff will be delayed. Due to the large amount of runoff that will be produced when a heavy rainfall falls, the runoff series will show a great jump. At this period, obvious runoff will be produced directly on karst slopes because soil moisture is absorbed and saturated in a short time due to the large amount of rainfall, and its response to rainfall is timely with the consistent variation relationship. On the contrary, the runoff series may jump negatively due to the lack of rainfall when the watershed suffers from years of rare drought, but the positional correlation between the two is positive. The effect of rainfall on runoff can be influenced by human activities, such as soil and water conservation, which may play an important role in reducing runoff. However, the role of soil and water conservation will become small or ineffective when encountering heavy rain or rainstorm. The effect of rainfall on runoff changes will change the relationship between rainfall and runoff because of different patterns, intensity or frequency when raining. Temporary water intake by human activities can also alter runoff, thereby resulting in different time effects of advance, synchronization and lag.

*6.2. Multi-Scale Effects of Evaporation on Runoff Changes*

In the process of rainfall, evaporation exerts a minimal effect on runoff but has a great impact on the water storage capacity of the basin before rainfall. The greater the evaporation intensity, the smaller the soil water content before the rain, which increases the infiltration loss of rainfall and reduces the small-diameter flow. This study has supported the previous conclusion in annual scale that the effect of evaporation on runoff change was significantly enhanced, showing a great contribution of 10–90% [32], but there were some new discoveries during the year. The effect of evaporation on runoff was only in the high-frequency scale in summer and the 6-year scale in winter. In other seasons or scales, most hysteresis effects with a few synchronous relationships in the time domain have shown at different time scales, which indicates that runoff changes are affected by evaporation. The evaporation is larger in summer; hence, short-term evaporation has a significant impact on runoff changes, which results in the changes in runoff lagging behind evaporation. The effect of runoff ahead of evaporation has been virtually masked by rainfall and human activities. The runoff changes are greatly influenced by abrupt rainfall and human activities, whilst evaporation shows a continuous stable process. Rainfall burst or human activities will contribute to the changes in the underlying surface of the watershed, which directly alters the evaporation conditions and volume that will cause the time dislocation in different time domains. The strong disturbance of human activities on runoff will directly lead to relatively stable and persistent evaporation lagging behind the change in runoff.

The essence of changing runoff by evaporation is to reduce the recharge of runoff and increase the evaporation of the river surface. In addition, evaporation shows a high impact on runoff change also because of the influence of the subtropical monsoon climate, abundant light and heat resources in Southwest China. The influence of evaporation on runoff varies obviously in different periods, which is mainly affected by the light, temperature, heat, climate and water content of underlying surface. However, runoff changes are affected not only by evaporation but also by rainfall and human activities, which makes it impossible for the evolution of runoff and evaporation to be completely consistent.

*6.3. Multi-Scale Effects of Temperature on Runoff Changes*

The influence of temperature on runoff is consistent with that of evaporation in both time and frequency domains and has the same multi-time-scale characteristics and time–frequency relationship. However, a negative correlation exists between temperature and runoff because an increase in temperature leads to the intensification of evaporation on slopes and rivers of the watershed and decreases air humidity, thus changing the runoff.

On monthly and annual scales, as well as in summer, the effect of temperature on runoff is mostly ahead of schedule, whereas it is mainly lagging behind in spring, autumn and winter. Thus, the regulation of temperature on runoff is mainly reflected on the season

scale. In summer, the change in runoff is mainly caused by changing evaporation and increasing rainfall to recharge soil moisture, and hence its impact on runoff shows a longer lead time than that of rainfall and evaporation. In other seasons, human activities change runoff intensely because of the relatively minimal rainfall, which leads to the relative lag of temperature change. On annual scale, the temperature regulation effect before 2000 is relatively stable, which mainly changes the runoff by changing the roles of evaporation and rainfall, resulting in a leading effect. Overall, the inter-annual temperature regulation is gradually lost, and the temperature regulation during the year is relatively prominent, but this regulation remains affected by human activities.

## 7. Conclusions

In this study, the multi-scale influences of climate factors on runoff changes in the Yinjiang River watershed are identified by using wavelet analysis, and the evolution relationship of time and frequency between runoff changes and climatic factors is further revealed at different time scales. The main conclusions are as follows:

(1) All climatic factors exhibit a main cycle at 12-month time scales with runoff changes, but the main periodic bandwidth of rainfall on runoff changes is much wider than that of temperature and evaporation, indicating that rainfall is the main factor affecting runoff changes.

(2) In other cycles, the impact of rainfall on runoff changes is the interlacing phenomena with positive and negative, but the impact of temperature and evaporation on runoff change is mainly negative.

(3) The response of runoff to rainfall is timely in the high-energy region and the low-energy significant-correlation region and shows a positive correlation with a smaller phase angle, but it is slightly lagged at 16-month time scales, in which the runoff changes lag behind temperature and evaporation for 1–2 months.

(4) It has been found that there is a strong effect of rainfall over runoff but a lesser effect of temperature and evaporation over runoff.

(5) The interaction of rainfall with runoff changes at 6-year time scales across the entire period. The effect of evaporation on runoff changes is similar to that of temperature and exhibited obvious local characteristics, mainly at small cycles.

The study has revealed the evolution process of river runoff in typical karst basins and the interaction mechanism between river runoff and climatic factors on multiple time scales, providing theoretical inspiration for fully solving the regional water shortage and engineering water shortage problems in the karst areas of Guizhou Province.

**Author Contributions:** Conceptualization, methodology and validation, X.B. and S.W.; Software and data curation, L.W. and C.R.; Formal analysis, L.W. and C.L.; Investigation, L.W. and F.C.; Writing—original draft preparation, L.W., C.R. and S.Z. All authors have read and agreed to the published version of the manuscript.

**Funding:** This research work was supported jointly by the Western Light Cross-team Program of Chinese Academy of Sciences (No. xbzg-zdsys-202101), National Natural Science Foundation of China (No. 42077455 & No.42167032), Strategic Priority Research Program of the Chinese Academy of Sciences (No. XDB40000000 & No. XDA23060100), Guizhou Provincial Science and Technology Projects (No. 2022-198), High-level innovative talents in Guizhou Province (No. GCC[2022]015-1 & No. 2016-5648), Guizhou Provincial 2020 Science and Technology Subsidies (No. GZ2020SIG), Opening Fund of the State Key Laboratory of Environmental Geochemistry (No. SKLEG2021072001 & No. SKLEG2022206 & No. SKLEG2022208) and Doctoral Research Startup Fund Project of Tongren University (No. trxyDH2103).

**Institutional Review Board Statement:** Not applicable.

**Informed Consent Statement:** Not applicable.

**Data Availability Statement:** The data analyzed in this study are subject to the following licenses/ restrictions: The dataset can only be accessed from China Meteorological Data Sharing Service System, Karst Scientific Data Center and Guizhou Provincial Hydrology and Water Resources Bureau. Requests to access these datasets should be directed to jgywlh@gztrc.edu.cn.

**Acknowledgments:** We would like to thank all the authors and reviewers for their great guidance and help in writing this manuscript.

**Conflicts of Interest:** The authors declare no conflict of interest.

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
