# Peer review of "Identifying the Multi-Scale Influences of Climate Factors on Runoff Changes in a Typical Karst Watershed Using Wavelet Analysis"

_land, doi:10.3390/land11081284_

Round 1

Reviewer 1 Report

The paper presents results of research on the impacts of selected climatic elements on changes in river runoff in karst areas. In the study the wavelet analysis methods were used, and the Yinjiang River watershed in the north-eastern part of Guizhou Province, South-West China was taken as the case study. In my opinion, it is an interesting study, which deserves attention and the paper can be published in the journal. While I don’t have any specific remarks related to methodology, I would suggest some (mainly editorial and linguistic) improvements before the final acceptance of the manuscript for publication, including:

1. Please write “Pearson correlation coefficient” instead of “pearson correlation coefficient” (p. 2, l. 50), then “the Yinjiang River watershed” instead of “the Yinjiang river watershed” (p. 3, l. 130).

2. P. 3, l. 136: “(…) mean elevation of 1033 m.” – please add “above sea level” at the end of the sentence.

3. Figure 1: in sub-figure (b) please add “Elevation (m a.s.l.)”.

4. Please check the paper carefully for language errors.

5. You may also underline the practical value of your study in relation to the water management problems faced by Guizhou Province.

With regard to the above-mentioned remarks it is recommended to accept the paper for publication after minor corrections.

Author Response

Dear Editors and Reviewers:

First of all, thank you and reviewers for the extremely helpful comments concerning our manuscript entitled “Identifying the multi-scale influences of climate factors on runoff changes in a typical karst watershed using wavelet analysis” (Land-1828041). Those comments are all valuable and very helpful for revising and improving our paper, as well as the important guiding significance to the researches. Thank you for your patience and the reviewers for their valuable comments and advice. We have studied comments carefully and tried our best to revise the manuscript according to every single comment which made by the reviewers and the editors. We have marked every changes with highlight colors in this revised version (See the the updated “Manuscript (revision changes marked)).

Point-by-point responses to the reviewers’ comments are listed below this letter. According to the editors and reviewers' suggestions, we have systematically revised our Manuscript. We hope that the revised version of the manuscript is now acceptable for publication in your journal.

Thank you for your consideration!

Sincerely yours,

Luhua Wu

*Corresponding Author: Xiaoyong Bai

  • mail: baixiaoyong@vip.skleg.cn

====================================================Point-by-Point Responses to Editor and Reviewers Comments

====================================================

Thank you for the patience and careful work. Both the technical and grammatical

revisions have been made thoroughly in the updated PDF version of “Manuscript (revision changes marked)”. The point-by-point responses to the reviewer’s comments are listed as following:

Response to Reviewer 1 Comments

Comments: The paper presents results of research on the impacts of selected climatic elements on changes in river runoff in karst areas. In the study the wavelet analysis methods were used, and the Yinjiang River watershed in the north-eastern part of Guizhou Province, South-West China was taken as the case study. In my opinion, it is an interesting study, which deserves attention and the paper can be published in the journal. While I don’t have any specific remarks related to methodology, I would suggest some (mainly editorial and linguistic) improvements before the final acceptance of the manuscript for publication, including:

  1. Please write “Pearson correlation coefficient” instead of “pearson correlation coefficient” (p. 2, l. 50), then “the Yinjiang River watershed” instead of “the Yinjiang river watershed” (p. 3, l. 130).
  2. P. 3, l. 136: “(…) mean elevation of 1033 m.” – please add “above sea level” at the end of the sentence.
  3. Figure 1: in sub-figure (b) please add “Elevation (m a.s.l.)”.
  4. Please check the paper carefully for language errors.
  5. You may also underline the practical value of your study in relation to the water management problems faced by Guizhou Province.

With regard to the above-mentioned remarks it is recommended to accept the paper for publication after minor corrections.

Overall Response: Thank you very much for your significant comments and instructive advice, which is really important for us to modify and improve the manuscript and has significant guiding significance for our future work. Your valuable advice to expand this research to the whole world points out the direction for our future research. According to your nice suggestions, we tried our best to improve the manuscript and made some revises in the manuscript. Our point-by-point responses are marked in blue font, and the parts is marked in bold. We have marked every change with highlight color in the updated PDF version of “Manuscript (revision changes marked)”. For the all questions you mentioned, my responses and improvements are as follows.

Point-by-Point Responses to Each Question of Reviewer

Questions 1: Please write“Pearson correlation coefficient”instead of“pearson correlation coefficient” (p. 2, l. 50), then “the Yinjiang River watershed” instead of “the Yinjiang river watershed”(p. 3, l. 130).

Response: Thank you for your patience and careful work. We Have written “Pearson correlation coefficient” instead of “pearson correlation coefficient” and “the Yinjiang River watershed” instead of  “the Yinjiang river watershed”. Please see on line 52 in the updated PDF version of “Manuscript (revision changes marked)”.

Questions 2: P. 3, l. 136: “(…) mean elevation of 1033 m.” – please add “above sea level” at the end of the sentence.

Response: Thank you very much for your instructive advice, we have added the “above sea level” at the end of the sentence “(…) mean elevation of 1033 m”. Please see on line 142 in the updated PDF version of “Manuscript (revision changes marked)”.

Questions 3: Figure 1: in sub-figure (b) please add“Elevation (m a.s.l.)”.

Response: Thank you for your patience and careful work. It is true that there is unclear for the description of Figure. 1: in sub-figure (b). Now, we have added the “Elevation” in sub-figure (b). Please see on line 142-143 in the updated PDF version of “Manuscript (revision changes marked)” and the updated Figure 1b.

Questions 4: Please check the paper carefully for language errors.. 

Response: Thank you for your patience and careful work. We apologized for the poor language of our manuscript. Your suggestions made us aware of the language errors in our manuscript. Now, we have checked all the language errors carefully for our Manuscript. Please see on revision mark in the updated PDF version of “Manuscript (revision changes marked)”.

In additon, we have carefully checked and corrected the reference formats in the whole revised manuscript according to the requirements of the Journal. For example, the revision of journal name abbreviation and verification of interval symbols. Please see the section of ‘Reference’ in the updated PDF version of “Manuscript (revision changes marked)”. 

Questions 5: You may also underline the practical value of your study in relation to the water management problems faced by Guizhou Province. 

Response: Thank you for your instructive suggestions. We have rewritten the practical value of our study in Abstract ang Conclusions. The revised sentence in Abstract is “The study shed light on the main teleconnections between rainfall, evapotranspiration and surface runoff, which in turn might help attaining better management of water resources in typical karst watersheds”. Please see on line 30-32 in the updated PDF version of “Manuscript (revision changes marked)”. The revised sentence in Conclusions is “The study has revealed the evolution process of river runoff in typical karst basins and the interaction mechanism between river runoff and climatic factors on multiple time scales, providing theoretical inspiration for fully solving the regional water shortage and engineering water shortage problems in karst areas of Guizhou Province.”. Please see on line 859-862 in the updated PDF version of “Manuscript (revision changes marked)”. 

Reviewer 2 Report

The authors provided a very interesting draft regarding the multiscale analysis of climate factors on runoff in a typical karst environments.

The study is interesting and the topic is quite novel. However, there are many concerns regarding the paper and I suggest a major revision. Besides, I am giving major comments and I suggest the authors address them point by point.

Major comment 1: Among the climate factors considered, there is evaporation, which is quite confusing to me. Generally, hydrological studies consider potential evapotranspiration (which actually conveys both evaporation and transpiration). Why did the authors chose to use only evaporation here, and where does the data come from ?

Major comment 2: finally, considering the results, the authors does not provided any physical description/explanation of the specifics of their findings in relation with karst hydrology. The findings are well defined, but the reader is left still lacking why are the findings so specific to karst hydrosystems. Also, I suggest the authors make a shorst comparison of their findings with other classical hydrosystems.

Overall major comment: the result and discussion sections are extremely long, and should be shortened to the major findings. On the contrary, the methodology section is not well elaborated, and the differents steps and outcome are not clearly specified. I suggest the authors ameliorate this section and also provide a flowchart of their analysis.

I have many other comments left within the document attached, to help the authors improve the manuscript. I definitely believe the paper has potential for publication in the Land Journal. Congrats to the authors.

Author Response

Dear Editors and Reviewers:

First of all, thank you and reviewers for the extremely helpful comments concerning our manuscript entitled “Identifying the multi-scale influences of climate factors on runoff changes in a typical karst watershed using wavelet analysis” (Land-1828041). Those comments are all valuable and very helpful for revising and improving our paper, as well as the important guiding significance to the researches. Thank you for your patience and the reviewers for their valuable comments and advice. We have studied comments carefully and tried our best to revise the manuscript according to every single comment which made by the reviewers and the editors. We have marked every changes with highlight colors in this revised version (See the the updated “Manuscript (revision changes marked)).

Point-by-point responses to the reviewers’ comments are listed below this letter. According to the editors and reviewers' suggestions, we have systematically revised our Manuscript. We hope that the revised version of the manuscript is now acceptable for publication in your journal.

Thank you for your consideration!

Sincerely yours,

Luhua Wu

*Corresponding Author: Xiaoyong Bai

  • mail: baixiaoyong@vip.skleg.cn

====================================================Point-by-Point Responses to Editor and Reviewers Comments

====================================================

Thank you for the patience and careful work. Both the technical and grammatical

revisions have been made thoroughly in the updated PDF version of “Manuscript (revision changes marked)”. The point-by-point responses to the reviewer’s comments are listed as following:

Response to Reviewer 2 Comments

Comment: The authors provided a very interesting draft regarding the multiscale analysis of climate factors on runoff in a typical karst environments.

The study is interesting and the topic is quite novel. However, there are many concerns regarding the paper and I suggest a major revision. Besides, I am giving major comments and I suggest the authors address them point by point.

Major comment 1: Among the climate factors considered, there is evaporation, which is quite confusing to me. Generally, hydrological studies consider potential evapotranspiration (which actually conveys both evaporation and transpiration). Why did the authors chose to use only evaporation here, and where does the data come from?

Major comment 2: finally, considering the results, the authors does not provided any physical description/explanation of the specifics of their findings in relation with karst hydrology. The findings are well defined, but the reader is left still lacking why are the findings so specific to karst hydrosystems. Also, I suggest the authors make a shorst comparison of their findings with other classical hydrosystems.

Overall major comment: the result and discussion sections are extremely long, and should be shortened to the major findings. On the contrary, the methodology section is not well elaborated, and the differents steps and outcome are not clearly specified. I suggest the authors ameliorate this section and also provide a flowchart of their analysis.

I have many other comments left within the document attached, to help the authors improve the manuscript. I definitely believe the paper has potential for publication in the Land Journal. Congrats to the authors.

Overall Response: Thanks for your professional review and valuable advice on our paper. We feel sincerely thanks for your agreement with our paper. As you are concerned, there are several problems that need to be addressed. According to your nice suggestions, We will be happy to edit the text further, based on helpful comments from the reviewers. And we will try our best to improve the manuscript and make detailed revises in the manuscript. Our point-by-point responses are marked in blue font, and the parts is marked in bold. We have marked every change with highlight color in the updated PDF version of “Manuscript (revision changes marked)”. We appreciate for your warm work earnestly, and hope that the correction will meet with approval. The detailed corrections are listed below.

Questions 1: Among the climate factors considered, there is evaporation, which is quite confusing to me. Generally, hydrological studies consider potential evapotranspiration (which actually conveys both evaporation and transpiration). Why did the authors chose to use only evaporation here, and where does the data come from?

Response: Thank you for your valuable suggestions. That’s a very good and insightful idea, which is very important for us to improve the study. The purpose of this paper is to reveal the multi-scale impact of watershed climate factors on surface runoff, so evaporation (even water surface evaporation) is used, and evaporation includes water surface evaporation and vegetation transpiration. Vegetation transpiration mainly affects the soil water content of vegetation covered land, and has little impact on surface runoff. Water surface evaporation has a great impact on surface runoff, which may be one of the main reasons for the loss of surface runoff. Therefore, we consider evaporation rather than evapotranspiration (PET). We still thank you for your valuable suggestion, which is very important for our research and broadens our understanding. In the following research, we will consider your suggestion to use evapotranspiration (PET) to represent the amount of water lost from the studied water system.

As described in Section 3 materials, the monthly evaporation data (01-1984 to 12-2015), which is also derived from Guizhou Provincial Hydrology and Water Resources Bureau (http://www.gzswj.gov.cn/hydrology_gz_new/index.phtml), is measured by the evaporation dish of the hydrological observation station and represents the water evaporation of water surface or soil.

Questions 2: finally, considering the results, the authors does not provided any physical description/explanation of the specifics of their findings in relation with karst hydrology. The findings are well defined, but the reader is left still lacking why are the findings so specific to karst hydrosystems. Also, I suggest the authors make a shorst comparison of their findings with other classical hydrosystems.

Response: We agree. Thank you very much for your constructive comments. We have described and explained your valuable opinions in detail in the discussion section. We have provided some physical description/explanation of the specifics of important findings in relation with karst hydrology through the comparison between our results and previous researches. Nevertheless, we have also taken your suggestions into consideration in this revision and made careful revision and improvement.

The modified marks can be seen in “Section 6.1 Multi-time scale effects of rainfall on runoff in the updated PDF version of “Manuscript (revision changes marked)”.

Questions 3: the result and discussion sections are extremely long, and should be shortened to the major findings. On the contrary, the methodology section is not well elaborated, and the differents steps and outcome are not clearly specified. I suggest the authors ameliorate this section and also provide a flowchart of their analysis..

Response: Thank you very much for your constructive comments. Your professional suggestions greatly improve the logic, readability and understandability of our manuscript.

(1) We have condensed and summarized the result and discussion sections of the manuscript, and have presented the latest and most important findings. The modified marks can be seen in the result and discussion sections in the updated PDF version of “Manuscript (revision changes marked)”.

(2) After careful consideration and conception, We have supplemented the specific formula and parameter description of each analysis method and added a diagram (flowchart) of the different steps of the analysis. Please see in “Section 4 Methodology” in the updated PDF version of “Manuscript (revision changes marked)”.

Point-by-Point Responses to Other Comments Left by Reviewer within the Document Attached

We agree and have recognized that all your comments are valuable and constructive, which gives us good scientific inspiration and promotes the improvement and perfection of our manuscript. We have tried our best to revise the manuscript according to each single comment. For the questions you mentioned, my specific responses and improvements can be found from questions 1 to 24 as follows.

Questions 1:  I suggest using the term“teleconnection”.

Response: Thank you for your careful review and valuable suggestions. We are grateful for the comment. The statement has been corrected using the term “teleconnection” now. The modified marks can be seen on line 16 in the updated PDF version of “Manuscript (revision changes marked)”.

Questions 2: Provide a quantitative measure. 

Response: We agree. Thank you for your patience and careful work. We have provided a quantitative measure about the time scale.The modified marks can be seen on line 22-23 in the updated PDF version of “Manuscript (revision changes marked)”.

Questions 3: Are there quantitative measures of the contributions of rainfall and evapotranspiration on surface runoff ? If yes, please report them..

Response: Thank you very much for your instructive advice, which is really important for us to improve our manuscript. In fact, we have been thinking for a long time about your suggestions. For the quantitative measurement of the contribution of rainfall and Evapotranspiration to surface runoff, we have completed the research in the previously published papers, and the research results have been published in the 《Science of the Total Environment》“Wu L, Wang S, Bai X, et al. Quantitative assessment of the impacts of climate change and human activities on runoff change in a typical karst watershed, SW China[J]. Science of The Total Environment, 2017, 601-602:1449-1465”. The previous research results showed that the contribution of precipitation to runoff change was 50%-60% and was considered high and stable. The contribution of evaporation to runoff change was10%-90% and was variable with a positiveor negative effects.The purpose of this paper is to reveal the multi-scale impact of climate factors on the change of surface runoff on different time scales. Thank you for your valuable and professional advice.

Questions 4: I would rather suggest stating that the study sheds light on the main teleconnections between rainfall, evapotranspiration and surface runoff, which in turn might help attaining better management of water resources in typical karst watersheds.

Response: Thank you for your valuable suggestions. Your finishing touch extremely accurately illustrates the research value of our manuscript. After careful consideration, we are very happy to decide to use your current statement to replace our previous sentence. Please see on line 30-32 in the updated PDF version of “Manuscript (revision changes marked)”.

Questions 5: Regarding how surface runoff is affected by soil surface conditions, there are three major references in the West African Sahel that investigate the topic (through measurements carried out under natural rainfall conditions over 6-10 years). Perhaps the authors can consider adding them here:

  1. Yonaba, R., Biaou, A.C., Koïta, M., Tazen, F., Mounirou, L.A., Zouré, C.O., Queloz, P., Karambiri, H., Yacouba, H., 2021. A dynamic land use/land cover input helps in picturing the Sahelian paradox: Assessing variability and attribution of changes in surface runoff in a Sahelian watershed. Science of The Total Environment 757, 143792. https://doi.org/10.1016/j.scitotenv.2020.143792.

Response: Thank you for your instructive suggestions. We have learned and quoted this excellent literature in the revised manuscript. Please see on line 39-40 in the updated PDF version of “Manuscript (revision changes marked)”.

Questions 6: Again, the study mentioned below carried this analysis. Yonaba, R., Biaou, A.C., Koïta, M., Tazen, F., Mounirou, L.A., Zouré, C.O., Queloz, P., Karambiri, H., Yacouba, H., 2021. A dynamic land use/land cover input helps in picturing the Sahelian paradox: Assessing variability and attribution of changes in surface runoff in a Sahelian watershed. Science of The Total Environment 757, 143792.https://doi.org/10.1016/j.scitotenv.2020.143792

Response: Thank you for your careful review. We have learned and quoted this excellent literature in the revised manuscript. Please see on line 51 in the updated PDF version of “Manuscript (revision changes marked)”.

Questions 7: Please move the following section to a new paragraph.

Response: We sincerely appreciate your valuable comment. We have moved the following section to a new paragraph. Please see on line 90 in the updated PDF version of “Manuscript (revision changes marked)”.

Questions 8: There are various application outside China, that can be mentioned as well: For example, in West Africa and Central Africa: https://dx.doi.org/10.1016/j.gloplacha.2019.04.003

Response: That’s a very good and insightful idea, which is very important for us to improve the paper. We have learned and quoted this excellent literature in the revised manuscript. Please see on line 102-104 in the updated PDF version of “Manuscript (revision changes marked)”.

Questions 9: At this point, the authors should be clear regarding the use of the term runoff, which is broad: is it surface runoff? Total runoff (i.e. water yield)?

Response:Thank you for your instructive suggestions. Your suggestions made us aware of the format errors in our references. We have careful rechecked and corrected this point about the use of the term runoff. It is surface runoff in our manuscript. Please see on line 126 in the updated PDF version of “Manuscript (revision changes marked)”.

Comments 10: Why is there a focus on evaporation? Actually where does the evaporation data come from? Does potential evapotranspiration (PET) come as better

factor to express the water withdrawal from the hydrosystem studied?

Response: Thank you for your valuable suggestions. As described in Section 3 materials, the monthly evaporation data (01-1984 to 12-2015), which is also derived from Guizhou Provincial Hydrology and Water Resources Bureau (http://www.gzswj.gov.cn/hydrology_gz_new/index.phtml), is measured by the evaporation dish of the hydrological observation station and represents the water evaporation of water surface or soil. The purpose of this paper is to reveal the multi-scale impact of watershed climate factors on surface runoff, so evaporation (even water surface evaporation) is used, and evaporation includes water surface evaporation and vegetation transpiration. Vegetation transpiration mainly affects the soil water content of vegetation covered land, and has little impact on surface runoff. Water surface evaporation has a great impact on surface runoff, which may be one of the main reasons for the loss of surface runoff. Therefore, we consider evaporation rather than evapotranspiration (PET). We still thank you for your valuable suggestion, which is very important for our research and broadens our understanding. In the following research, we will consider your suggestion to use evapotranspiration (PET) to represent the amount of water lost from the studied water system.

Comments 11: Panel (b): change “mian” to “main”

Response: Thank you for your patience and careful work. We have changed "mian" to "main" in Panel (b) of Figure 1. Please see on line 142-144 in the updated PDF version of “Manuscript (revision changes marked)” and the updated Figure 1b.

Comments 12: Panel (b): add the title “Elevation (masl)” for the range 439-2466

Response: Thank you for your patience and careful work. We have added the title “Elevation (m)” for the range 439-2466 in Panel (b) of Figure 1. Please see on line 142-143 in the updated PDF version of “Manuscript (revision changes marked)” and the updated Figure 1b.

Comments 13: Panel (b) and (c) : what are the sources for Elevation data (b) and lithology (c) ?

Response: As described in Section 3 materials. The lithology data is derived from the Karst Scientific Data Center (http://www.karstdata.cn/), Institute of geochemistry, Chinese Academy of Sciences. Here, We further added the source and spatial resolution of DEM data. The DEM data with a spatial resolution of 30 m was obtained from the International Scientific and Technical Data Mirror Site, Computer Network Information Center, Chinese Academy of Sciences, which could be downloaded from the Geospatial Data Cloud (http://www.gscloud.cn) (http://www.gscloud.cn). Please see on line 172-177 in the updated PDF version of “Manuscript (revision changes marked)”.

Comments 14: I highly recommend the authors to add a diagram (flowchart) of the different steps of the analysis, which will also (ultimately) include small description of the expected outputs at each step of the analysis.

Response: Thank you very much for your constructive comments. Your professional suggestions greatly improve the logic, readability and understandability of our manuscript. After careful consideration and conception, we have added a diagram (flowchart) of the different steps of the analysis. Please see in “Section 4 Methodology” in the updated PDF version of “Manuscript (revision changes marked)”.

Comments 15: Also, the authors should specify some details of the wavelet transform analysis carried: the type of wavelet? Their order ? the confidence level (against red noise) ?

Response: We agree, Thank you very much for your constructive comments. We have tried our best to revise your proposal. We have specified some details of the wavelet transform analysis in Figure 9 to 16 about the type of wavelet, their order and the confidence level.

Comments 16: I recommend the authors to change the order of display: (a) should be rainfall, (b) temperature, (c) evaporation and (d) (surface?) runoff. Authors should also outline the similarity between rainfall/runoff variations and similarly, the similarity between temperature/evaporation.

Response: We agree. Thank you very much for your constructive comments. We have changed the order of display in Figure 3,4,5,6,7, and 8 according to your suggestions. According to your suggestion, we have adjusted the arrangement order of the diagram and found that the logical relationship of the diagram is more clear. Thank you for your rich and professional opinions

Comments 17: What seasons ? List them and make sure the title of each of these figures (3, 4, 5 and 6) contains the name of its corresponding season.

Response: We apologize for the confusion generated by the imprecise description of this sentence and sincerely hope that our logic is now easier to follow with this new revised sentence. The new revised sentence is “ As shown in Figure. 4 to 7, there are some synchronization characteristics on different time scales for the evolution characteristics of surface runoff and climate factors on the annual scale and four seasonal scales after scale segmentation by MRA.”. Please see on line 338-341 in the updated PDF version of “Manuscript (revision changes marked)”.

Comments 18: Similar comment regarding the order of the panels

Response: We agree. Thank you very much for your constructive comments. We have changed the order of display in Figure 3,4,5,6,7, and 8 according to your suggestion. According to your suggestion, we have adjusted the arrangement order of the diagram and found that the logical relationship of the diagram is more clear. Thank you for your rich and professional opinions

Comments 19: I would suggest using the term “wavelet power spectra”, which is much more common as far as I know.

Response: Thank you for your careful review and valuable suggestions. Your opinion is very good. We have considered it very carefully. This small paragraph is a brief description of how to interpret Figure 9. We think that these contents are too complicated and unnecessary, and it is not necessary for professional scientific researchers to carry out specific interpretation and description. Therefore, we have deleted the paragraph where this sentence is located so that this subsection can directly present the research results. Please see on line 374-382 in the updated PDF version of “Manuscript (revision changes marked)”.

Comments 20: How did the authors detect this breakpoint? It was not mentioned previously.

Response: Thank you for your patience and careful work. This breakpoint was detected by CWT in Figure 9:Month-Q at 8-16- month time scales in 1990. We have corrected the statement now. Please see on line 377 in the updated PDF version of “Manuscript (revision changes marked)”.

Comments 21: Since the fluctuation in SummerQ is outside the cone of influence, it should not be analyzed.

Response: Thank you for your instructive suggestions. We are very willing to adopt your suggestions. We have checked carefully according to your suggestions and deleted some unnecessary analysis.

Comments 22: Please indicate the confidence level of the significant fluctuations.

Response: We agree. Thank you very much for your constructive comments. We have tried our best to revise your proposal. We have specified some details of the wavelet transform analysis in Figure 9 to 16 about the confidence level.

Comments 23: Interesting discussion. However, there is two concerns to be discussed in my humble opinion.(1) Generally speaking, the authors found a large/strong effect of P over Q, and a lesser effect of T and E over Q. However, it seeps that T and E are

highly related. I even believe that T is a major causality factor of E, which is quite understandable considering the physical causes of E. The authors should stress on this fact which is quite well known, and the results of which are quite expected, even before carrying the analyses.

Response: Thank you very much for your constructive comments. We very much agree with your understanding and admire your profound and professional knowledge. Thank you for learning and communicating with you. As you said, evaporation is mainly controlled by temperature, which is a fact. Here, the main purpose of this paper is to reveal the multi-scale impact of rainfall, temperature and evaporation on surface runoff, which may not exclude the process of temperature controlling and affecting evaporation. We also explained in the discussion section of 6.3 Multi-time-scale effects of temperature on runoff that temperature mainly changes the runoff change by changing evaporation and increasing rainfall to replenish soil moisture. Thank you very much for your friendly guidance. We will carefully study your suggestions and enlightenments and apply them in the following study and research.

Comments 24: I also do not really understand why the authors consider Evaporation (E) in this study. In my opinion, PET (potential Evapotranspiration) which conveys both evaporation and transpiration should be considered. However, I do no consider this as a major flaw and I would like the authors to provide strong motivations regarding their knowledge of the context and the physical basis behind such choice.

Response: Thank you for your valuable suggestions. We agree with your suggestion very much. However, as the purpose of this paper is to reveal the multi-scale impact of watershed climate factors on surface runoff, so evaporation (even water surface evaporation) is used, and evaporation includes water surface evaporation and vegetation transpiration. Vegetation transpiration mainly affects the soil water content of vegetation covered land, and has little impact on surface runoff. Water surface evaporation has a great impact on surface runoff, which may be one of the main reasons for the loss of surface runoff. Therefore, we consider evaporation rather than evapotranspiration (PET). We still thank you for your valuable suggestion, which is very important for our research and broadens our understanding. In the following research, we will consider your suggestion to use evapotranspiration (PET) to represent the amount of water lost from the studied water system.

Round 2

Reviewer 2 Report

Dear authors,

After going through the revised version of your manuscript, I have the feeling that all of my comments and concerns have been thoroughly addressed. I do not have further comments, beside conducting a careful proofread for English grammar fixes.

Therefore I do recommend accepting your paper. Congrats.